# Glycine, serine and threonine metabolism confounds efficacy of complement-mediated killing

Zhi-xue Cheng[1,2,7], Chang Guo [1,7], Zhuang-gui Chen[3,7], Tian-ci Yang [4], Jian-ying Zhang[5], Jie Wang[1], Jia-xin Zhu[3], Dan Li[1], Tian-tuo Zhang[3], Hui Li[1,2,6], Bo Peng[1,2,6] & Xuan-xian Peng[1,2,6]

Serum resistance is a poorly understood but common trait of some difficult-to-treat pathogenic strains of bacteria. Here, we report that glycine, serine and threonine catabolic pathway is down-regulated in serum-resistant *Escherichia coli*, whereas exogenous glycine reverts the serum resistance and effectively potentiates serum to eliminate clinically-relevant bacterial pathogens in vitro and in vivo. We find that exogenous glycine increases the formation of membrane attack complex on bacterial membrane through two previously unrecognized regulations: 1) glycine negatively and positively regulates metabolic flux to purine biosynthesis and Krebs cycle, respectively. 2) α-Ketoglutarate inhibits adenosine triphosphate synthase, which in together promote the formation of cAMP/CRP regulon to increase the expression of complement-binding proteins HtrE, NfrA, and YhcD. The results could lead to effective strategies for managing the infection with serum-resistant bacteria, an especially valuable approach for treating individuals with weak acquired immunity but a normal complement system.

[1] Center for Proteomics and Metabolomics, State Key Laboratory of Bio-Control, School of Life Sciences, Sun Yat-sen University, University City, Guangzhou 510006, People's Republic of China. [2] Laboratory for Marine Biology and Biotechnology, Marine Fisheries Science and Food Production Processes, Qingdao National Laboratory for Marine Science and Technology, Qingdao 266071, China. [3] Third Affiliated Hospital of Sun Yat-sen University, Guangzhou 510630, People's Republic of China. [4] Zhongshan Hospital of Xiamen University, Xiamen 361004, People's Republic of China. [5] Henan Academy of Medical and Pharmaceutical Sciences, Zhengzhou University, Zhengzhou, Henan 450052, People's Republic of China. [6] Southern Marine Science and Engineering Guangdong Laboratory (Zhuhai), Zhuhai 519000, China. [7] These authors contributed equally: Zhi-xue Cheng, Chang Guo, Zhuang-gui Chen. Correspondence and requests for materials should be addressed to T.-t.Z. (email: zhtituli@163.com) or to H.L. (email: lihui32@sysu.edu.cn) or to B.P. (email: pengb26@sysu.edu.cn) or to X.-x.P. (email: pxuanx@sysu.edu.cn)

The human complement system acts either independently or together with the immune system to kill bacteria in the bloodstream and/or on mucosal surfaces[1,2]. However, a significant number of both Gram-negative and Gram-positive bacteria survive and proliferate in human blood that has a fully functional complement system. These pathogens are therefore referred to as serum-resistant bacteria, and they are associated with potentially serious systemic infections[3–5]. Serum resistance has been extensively investigated, but the mechanism remains poorly understood at the molecular level. Most studies attribute the serum resistance of Gram-negative bacteria to bacterial outer membrane proteins, lipopolysaccharide (LPS) and its antibodies, and capsular polysaccharide in inhibiting complement system activation[6–11]. Some outer membrane proteins bind complement-inhibiting factors, such as OmpA binding with C4b-binding protein, preventing the efficient formation of membrane attack complexes (MACs)[12–15], while OmpC mutations increase bacterial survival in serum by inhibiting classical C1q and anti-OmpC antibody-dependent cell killing[16]. The serum resistome of E. coli EC958 comprised of 56 genes, where a major proportion of the genes encode membrane proteins or factors involving in LPS biosynthesis[17]. LPS is necessary for the survival of E. coli in the blood[18]. Immunoglobulin G to LPS confers serum resistance through shielding the bacteria from other antibodies that can induce complement-mediated killing[10,19,20]. The presence of capsular polysaccharide is correlated with serum resistance in bacteria[11]. Moreover, metabolism also plays roles in serum susceptibility, e.g., ferric uptake regulator (Fur) confers serum resistance[21,22] and $O_2$ consumption, adenine uptake, and glucose consumption increase are characteristic metabolic features when bacteria were exposed to serum[23,24].

These findings provide a clue to the molecular basis of serum resistance in Gram-negative bacteria[6–10,18–22]. However, they do not provide insight into the metabolic state of serum-resistant bacteria, nor do they suggest promising strategies for managing infections with serum-resistant pathogens. To address this knowledge gap, our lab recently analyzed the metabolome of antibiotic-resistant bacteria. The results led us to propose that exogenous metabolite(s) might constitute a perturbation sufficient to restore antibiotic-sensitivity to multidrug-resistant bacteria[25–29]. We also showed there is a link between bacterial serum resistance with metabolism[29,30]. In the present study, we describe a framework based on functional metabolomics and metabolite perturbation, that might lead to a novel approach for controlling and/or preventing acute infection with serum-resistant and/or antibiotic-resistant pathogens.

## Results

**Glycine catabolism is reduced in serum-survival E. coli.** Comprehensive metabolomics provides information about the metabolic state of a cell or an organism, and how that state changes in different environmental contexts[25,31]. We first showed that non-pathogenic E. coli K12 was killed by serum but not by heat-inactivated (HI) serum and control without serum (Supplementary Fig. 1a, b). Then metabolome profiling was performed in these serum-treated bacteria as shown in (Supplementary Fig. 1c–e). In serum-survival E. coli K12 cells, 59 metabolites were present at significantly different concentrations than the control without serum ($p < 0.01$; Fig. 1a). Of the 59 metabolites, 44 were present at lower abundance in the presence of serum and 15 were present at higher abundance (Fig. 1b). A web-based metabolomics pathway analysis tool, MetPA, was used to evaluate the potential impact of differential metabolite abundance on metabolic pathways in the bacterial strain. This analysis identified 8 pathways differentially-expressed in E. coli K12 in the presence of serum,

with the largest predicted differential effect involving glycine, serine and threonine metabolism, with secondary and tertiary impact on alanine, aspartate and glutamate metabolism and the Krebs cycle (TCA cycle), respectively (Fig. 1c, d). Hierarchical clustering analysis identified glycine, serine, and threonine as three of the critically down-regulated metabolites (Fig. 1e; Supplementary Fig. 2), substantiating glycine, serine and threonine metabolism is the most significantly altered pathway in serum-survival E. coli K12. The same analysis was performed for cells exposed to HI serum, displaying different signature and including increased glycine, serine, succinate, and fumarate (Supplementary Fig. 3).

qRT-PCR was used to demonstrate whether the reduced concentrations of glycine, serine, and threonine are linked to their catabolism or/and glycine transport. The expression of glyA, kbl, sdaA, sdaB, aceE, aceF, and gltA mediating glycine flux to the TCA cycle was decreased, whereas itaE and purD mediating glycine to threonine and glycinamide ribonucleotide, respectively, was elevated when exposed to serum (Fig. 1f). Meanwhile, serum repressed the transcription of glycine transporter gene, cycA (Fig. 1g). Thus, downregulation of glycine, serine and threonine catabolism to the TCA cycle is associated with active serum exposure, which is partly attributed to the decreased expression of glycine transporter. The further experiment demonstrated higher viability after re-exposure to serum (Fig. 1h).

Eight serum-resistant and eight serum-sensitive clinical isolates of E. coli were selected (Supplementary Fig. 4). Their metabolic profiles also show a lower abundance of glycine, serine and threonine in serum-resistant strains than in serum-sensitive strains (Fig. 1i). Thus, it is possible that low abundance of glycine, serine, and/or threonine could serve as a biomarker of serum-resistance.

**Glycine, serine, and threonine restore serum sensitivity.** We hypothesized that uptake of an exogenous metabolite might perturb the metabolome of a serum-resistant bacterium and thereby restore sensitivity to killing by serum. To demonstrate this, exogenous glycine, serine, or threonine was added to a final concentration of 100 mM and cell viability was measured. The viability in the presence of serum was reduced by 85.0-, 6.8-, or 8.9-fold, respectively (Fig. 2a). Exogenous glycine also significantly reduced viability of serum-susceptible strains E. coli K12, Y5, and serum-resistant strains Y10, Y17 in the presence of serum, while no killing was observed in the controls (Supplementary Fig. 5a) and the effect was dependent on glycine dose (Fig. 2b), serum concentration (Fig. 2c), and time (Fig. 2d, e). When serum was treated by anti-C3, EDTA, or EGTA, glycine failed to potentiate serum to kill bacteria (Fig. 2f; Supplementary Fig. 5b). Survival of E. coli Y17 was independent of the presence of serum during a 30-minute incubation, while under similar conditions, E. coli K12 showed approximately 65% survival. However, in the presence of serum and glycine, Y17 cells lost viability over time, with less than 1% survival after 6 h (Fig. 2d, e). Exogenous glycine did not potentiate ampicillin or gentamicin to kill bacteria under the same condition to exclude the possibility that exogenous glycine impairs bacterial cell wall integrity as reported in Gram-positive bacteria[32] (Supplementary Fig. 5c), even if the integrity is key to bacterial survival in the presence of antibiotics[33]. In addition, sensitivity to serum was not restored by incubation in media containing various other amino acids and metabolites (Supplementary Fig. 5d). These data show that glycine, serine, or threonine specifically potentiates the ability of serum to kill strains of E. coli that are otherwise resistant to such killing.

**Effect of exogenous glycine on metabolic state.** As suggested earlier, it is possible that serum-resistant could be reverted to

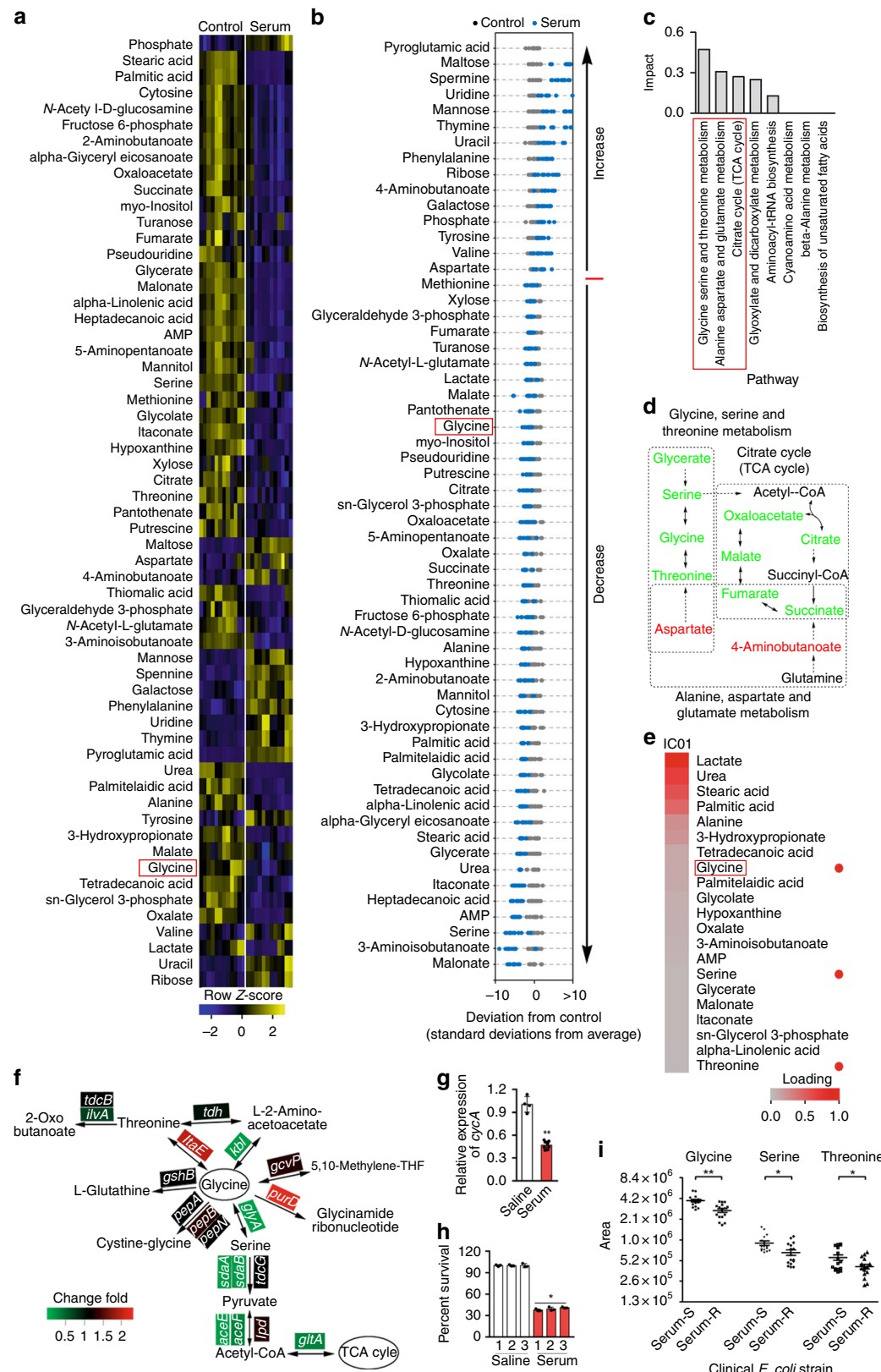

serum-susceptible based on the reprogramming metabolomes[31]. To test this, glycine was selected as a metabolome-reprogramming metabolite. Therefore, metabolomic profiles of 73 metabolites were compared in *E. coli* K12, as described above (Fig. 1a–e; Supplementary Fig. 1c–e), using bacteria grown in media or in media plus glycine, serum, or both (Supplementary

Figs. 6 and 7a–c). As compared to the control group, metabolites belonging to glycine, serine and threonine metabolism and to TCA cycle were lower in serum-exposed bacterial cells but increased when exposed to serum plus glycine except for threonine, which was decreased (Fig. 3a; Supplementary Figs. 7d and 7e). However, when serum plus glycine group was compared to

**Fig. 1** Serum-sensitive and -resistant bacteria have distinct metabolism. **a** Heat map showing relative abundance of 59 significantly differential metabolites in *E. coli* K12 in the absence (control) or presence (serum) of serum, as indicated. Heat map scale (blue to yellow, low to high abundance) is shown below data ($n = 6$). **b** Z-scores (standard deviation from average) corresponding to data in (**a**). **c** Pathway enrichment of significantly differential metabolites. Red box highlights the first three of most impacted pathways. **d**, Pathway interconnections of the first three most impacted pathways. The change of metabolite abundance is indicated as follows: black, no change; red, upregulation; green, downregulation; gray, not detected. **e** Hierarchical clustering of the decreased abundance of metabolites in serum-treated samples. Heat map scale (gray to red; low to high loading) is shown below data. Red box highlights glycine and red dot highlights glycine, serine and threonine metabolism. **f**, **g** qRT-PCR for the expression of glycine catabolism genes (**f**) and glycine transporter, *cycA* (**g**) in the presence or absence of 100 μL serum ($n = 3$). **h** Effect of repeated killing by serum on percent survival of *E. coli* K12. *E. coli* K12 was treated with serum (1) for selection of survivors, which was treated by the other two round of serum treatments (2 and 3). Percent of survival was calculated ($n = 3$). **i** Scatter plots showing a normalized abundance of glycine, serine, and threonine in eight serum-susceptible (Serum-S) and eight serum-resistant (Serum-R) clinical *E. coli* strains ($n = 8$). Results (**g–i**) are displayed as mean ± SEM, and significant difference is identified (*$p < 0.05$; **$p < 0.01$) as determined by two-tailed Student's *t* test. See also Supplementary Figs. 1–4

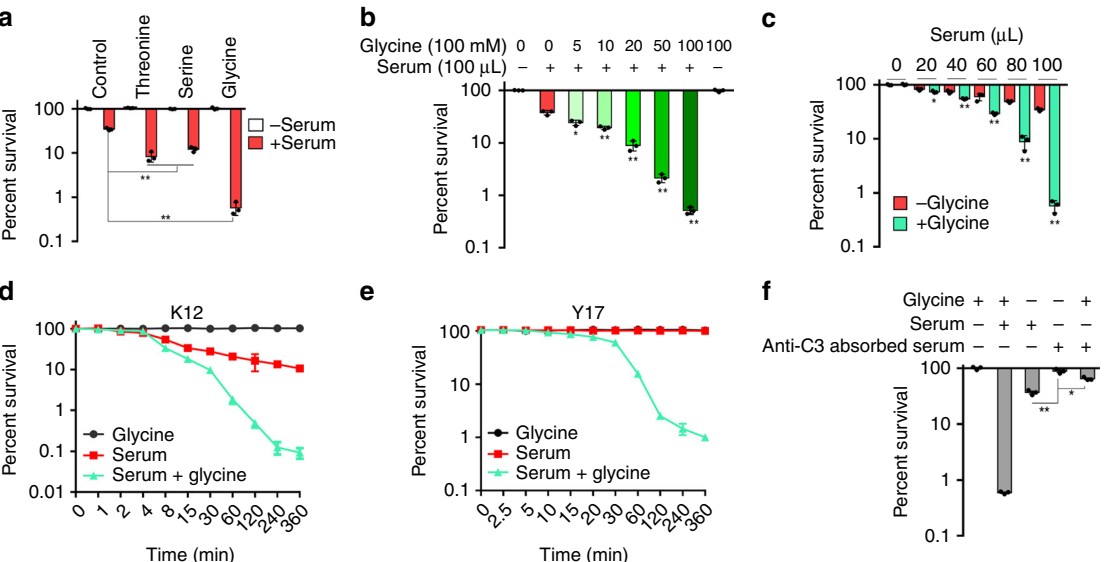

**Fig. 2** Glycine increases the susceptibility of *E. coli* to serum. **a** Percent survival of *E. coli* K12 incubated with 100 μL serum in the presence or absence of 100 mM glycine, serine, or threonine ($n = 3$). **b** Synergetic effects of 100 μL serum and glycine on viability of *E. coli* K12 were measured in a glycine dose-dependent manner (0–100 mM) ($n = 3$). **c** Percent survival of *E. coli* K12 incubated with 100 mM glycine plus serum (0–100 μL) or without glycine ($n = 3$). **d**, **e** Percent survival of *E. coli* K12 (**d**) and Y17 (**e**) in the presence of 100 mM glycine or/and 100 μL serum for the indicated length of time ($n = 3$). **f** Percent survival of *E. coli* K12 in the presence of 100 mM glycine or/and 100 μL serum or anti-C3 absorbed serum for 2 h ($n = 3$). Results are displayed as mean ± SEM (**a–f**), and significant differences are identified (*$p < 0.05$, **$p < 0.01$) as determined by two-tailed Student's *t* test (**a–c**, **f**). See also Supplementary Fig. 5

glycine group, the abundance of glycine and threonine was reduced, while the abundance of succinate, fumarate, malate, and oxaloacetate in the TCA cycle was increased, suggesting serum drives glycine flux to the TCA cycle (Supplementary Fig. 8). Therefore, these results suggest that serum plus glycine triggers a metabolic change leading to increased production of NADH via the TCA cycle (Fig. 3b).

NADH would increase the membrane potential, thus potentiating cell death in the presence of serum plus glycine. In fact, we observed that exogenous glycine or serum correlates with an increase in membrane potential in *E. coli* K12 and Y17, with a synergistic effect of serum plus glycine relative to controls (Fig. 3c). The ionophore carbonylcyanide m-chlorophenyl hydrazone (CCCP) abolished the synergistic increase in membrane potential (Fig. 3c) and bacterial percent survival (Fig. 3d) in *E. coli* K12 or Y17 cells exposed to serum plus glycine. These results suggest that increased membrane potential is crucial for the synergistic effect of serum and glycine to revert serum resistance. Interestingly, membrane potential is significantly higher in *E. coli* K12 than Y17, and a differential synergistic effect of serum and glycine on membrane potential is found for the two bacterial strains (Fig. 3c). The results are consistent with

the finding that *E. coli* K12 is more susceptible to the bactericidal activity of serum than Y17, and thus verifies the synergistic effect of serum plus glycine on survival.

With regard to the importance of membrane potential in serum-dependent cell death, it may relate to the recruitment of positively-charged C3 to the negatively-charged cell surface[34–36]. When membrane potential is sufficiently high, protons and/or electrons are transported across the membrane, which drives the synthesis of ATP, a reaction catalyzed by ATP synthase. Consistent with this, a defect in ATP synthase increases susceptibility to serum, which likely correlates with increased membrane potential (Fig. 3e; Supplementary Fig. 9).

We further validate the finding that exogenous glycine stimulates the TCA cycle using NTFD (non-targeted tracer fate detection)[37,38] (Fig. 3f; Supplementary Tables 1 and 2). Furthermore, qRT-PCR showed that the expression of genes mediating glycine catabolism was increased (Fig. 3g). These results indicate that exogenous glycine stimulates the TCA cycle.

Several other metabolites, including alanine, fructose, glucose, and glutamine stimulate the TCA cycle and increase membrane potential (Supplementary Fig. 10a)[25–27]; however, these metabolites did not decrease survival in the presence of serum

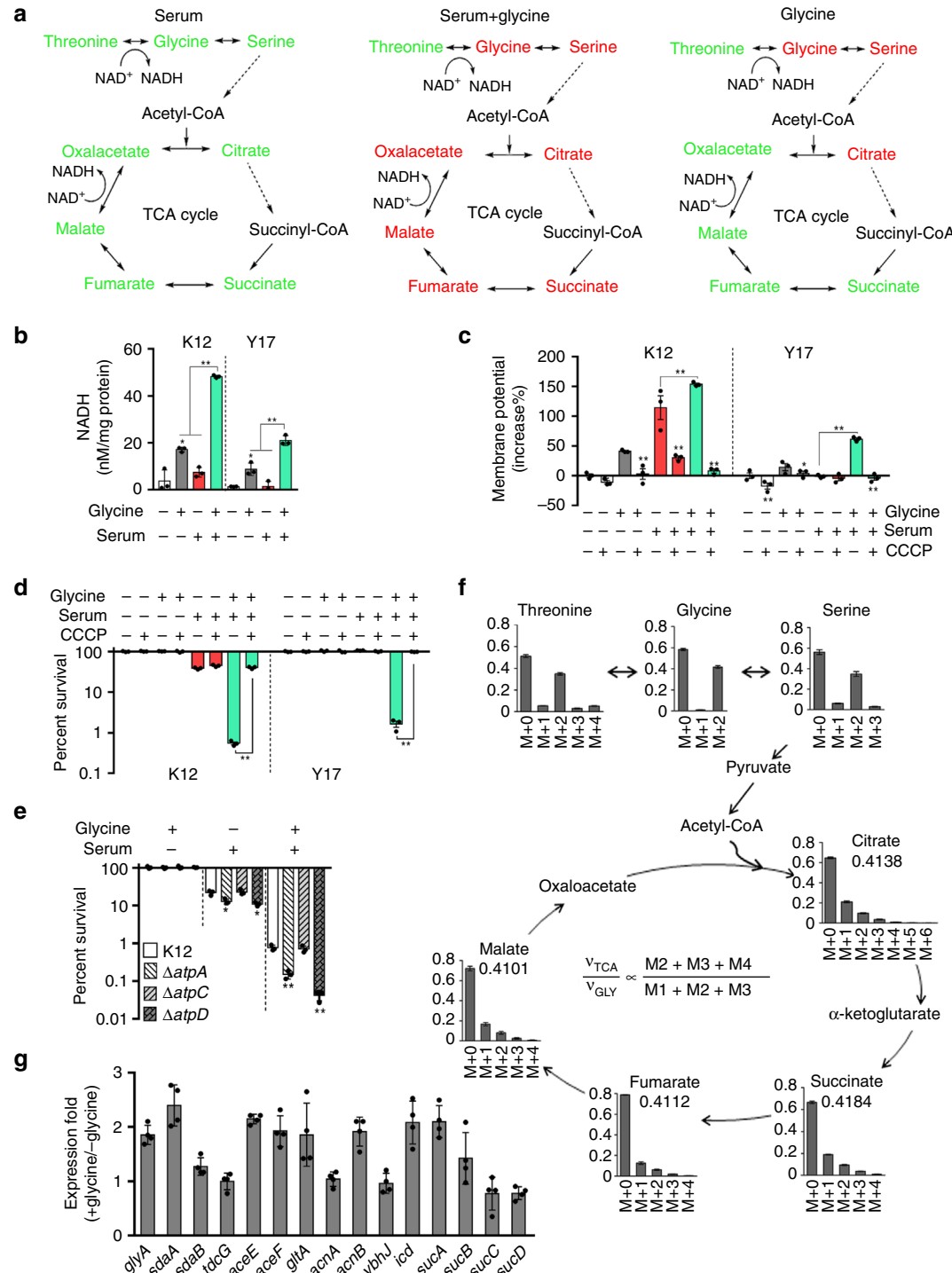

**Fig. 3** TCA cycle-triggered membrane depolarization. **a** Summary of metabolic flux through the TCA cycle in the presence of serum, serum plus glycine, or glycine. Change in metabolite abundance is indicated as follows: red, upregulation; green, down-regulation; gray, not detected. NAD+, nicotinamide adenine dinucleotide; NADH, reduced nicotinamide adenine dinucleotide ($n = 6$). **b** NADH concentration in *E. coli* K12 and Y17 in the presence of 100 μL serum, 100 mM glycine or both for 2 h ($n = 3$). **c, d** Membrane potential (**c**) and percent survival (**d**) of *E. coli* K12 and Y17 treated with 100 μL serum for 2 h in presence or absence of 100 mM glycine or/and CCCP ($n = 3$). For membrane potential, 50,000 cells were recorded with forwarding versus side scatter and were gated before data acquisition. **e** Percent survival of *E. coli*, Δ*atpA*, Δ*atpC*, and Δ*atpD* in the presence of 100 mM glycine, 100 μL serum, or both for 2 h ($n = 3$). **f** Mass isotopomer distributions for $^{13}$C labeled glycine detected in a nontargeted manner in the presence of 100 mM glycine. The relative flux for that metabolite in the TCA cycle ($\upsilon_{TCA}/\upsilon_{GLY}$) is defined (M2 + M3 + M4)/(M1 + M2 + M3) ratio, where $\upsilon_{TCA}$ refers to the turnover of a particular metabolite pool and $\upsilon_{GLY}$ refers to the flux of glycine carbon atoms to the TCA cycle. **g** qRT-PCR for relative expression of genes contributed to the flux from serine to the upper TCA cycle in the presence or absence of 100 mM glycine ($n = 3$). Results are displayed as mean ± SEM (**b–g**), and significant differences are identified (*$p < 0.05$, **$p < 0.01$) as determined by two-tailed Student's *t* test (**a–e**, **g**)

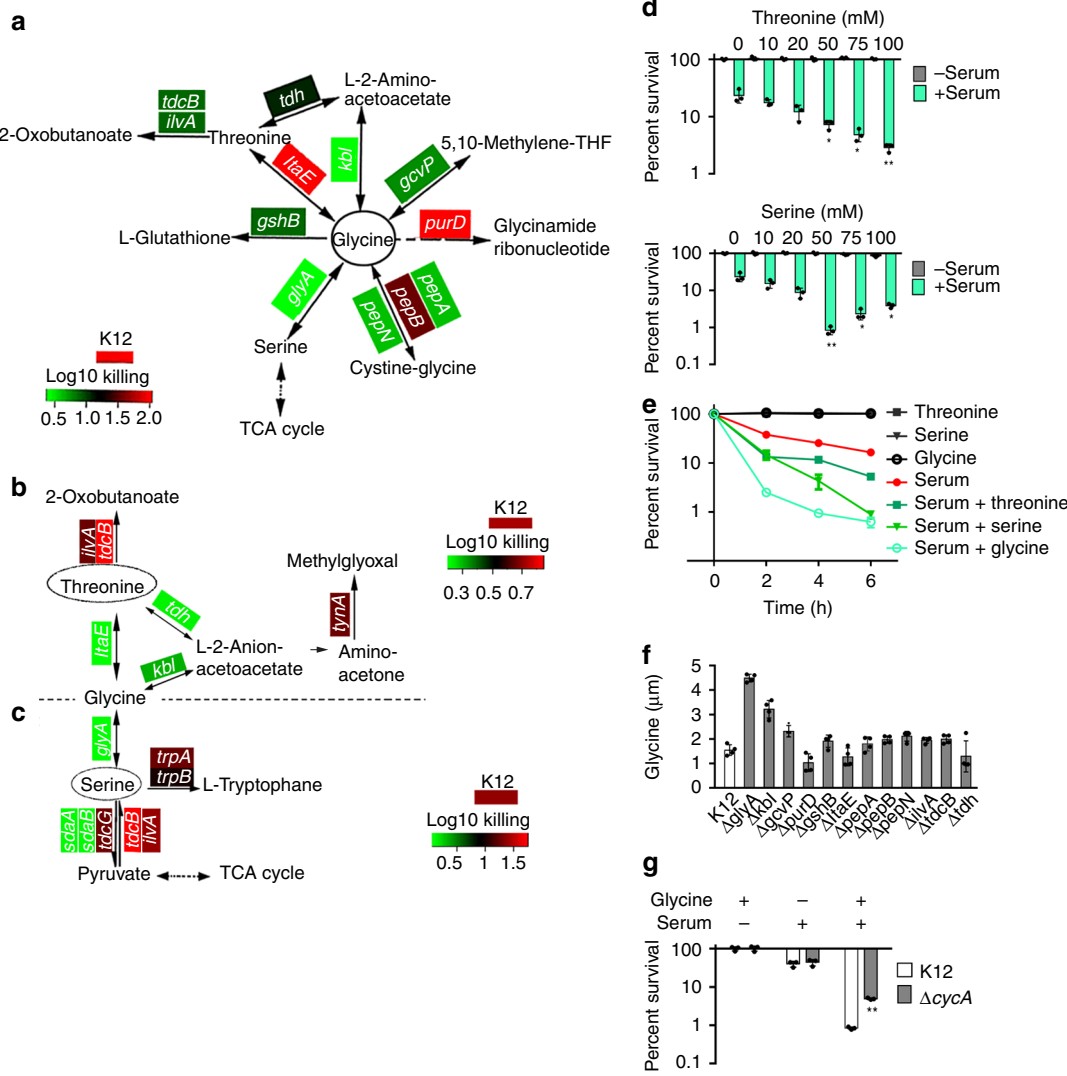

**Fig. 4** Glycine flux to the TCA cycle regulates bacterial serum susceptibility. **a–c** Survival of the *E. coli* mutants in the presence or absence of 100 μL serum plus 100 mM glycine (**a**), threonine (**b**) or serine (**c**). Heat map scale (blue to red) represents low to high killing without the indicated gene ($n = 3$). **d** Percent survival of *E. coli* K12 incubated with or without 100 μL serum plus different concentrations of serine (0–100 mM) or threonine (0–100 mM) ($n = 3$). **e** Percent survival of *E. coli* K12 incubated with 100 μL serum plus 50 mM glycine, serine, or threonine for different time points ($n = 3$). **f** Glycine level of the *E. coli* mutants as the same as (**a**) ($n = 4$). **g** Percent survival of *E. coli* K12 and Δ*cycA* in the presence of 100 μL serum, 100 mM glycine or both ($n = 3$). Results are displayed as mean ± SEM (**d–g**), and significant differences are identified (*$p < 0.05$, **$p < 0.01$) as determined by two-tailed Student's $t$ test (**d**, **f**, **g**). See also Supplementary Fig. 9

(Supplementary Fig. 5d). This suggests that other factors may also be involved in addition to membrane potential.

**Glycine flux to the TCA cycle is essential for promoting the serum killing**. One possible explanation is that glycine promotes the expression of glycine-responsive genes, which in turn mediates the killing. To test this possibility, 12 *E. coli* K12 single deletion mutants of glycine metabolism were cultured in the presence or absence of glycine plus serum. Increased viability was observed in Δ*glyA* and Δ*kbl* (Fig. 4a; Supplementary Fig. 9). These two genes encoded essential proteins to generate serine and threonine, respectively. Thus, these two proteins are required downstream of glycine to promote serum-dependent bacterial cell death, but they do not explain the mechanism of cell death.

Therefore, metabolic flux involving serine and threonine was explored. Genes that promote conversion of threonine to glycine

were required for the killing effect, but others were not (Fig. 4b; Supplementary Fig. 9). However, loss of genes, *sdaA*, and *sadB*, transferring serine to pyruvate, abrogated the killing (Fig. 4c; Supplementary Fig. 9). Thus, these three metabolites must flux to pyruvate to play a role, although interregulation among them existed, which is supported by the fact that the absence of *itaE* decreased the effect of glycine, and lower viability was detected in Δ*glyA* than Δ*tdcB* and Δ*ilvA* (Fig. 4a), suggesting it is required to flux threonine to glycine. Consistently, the viability of *E. coli* K12 cells was lower in the presence of serine than threonine at 6 h (Fig. 4d). When 50 mM serine or glycine was added, cell viability was equal at 6 h, but approximately four-fold fewer viable cells were detected than in the presence of the same concentration of threonine (Fig. 4e). We further demonstrated this through detection of glycine level in these mutants described in (Fig. 4a). The glycine concentrations were proportional to the percent survival of the mutants (Fig. 4f), indicating that glycine flux to the TCA cycle is essential. We also showed that loss of *cycA* increased

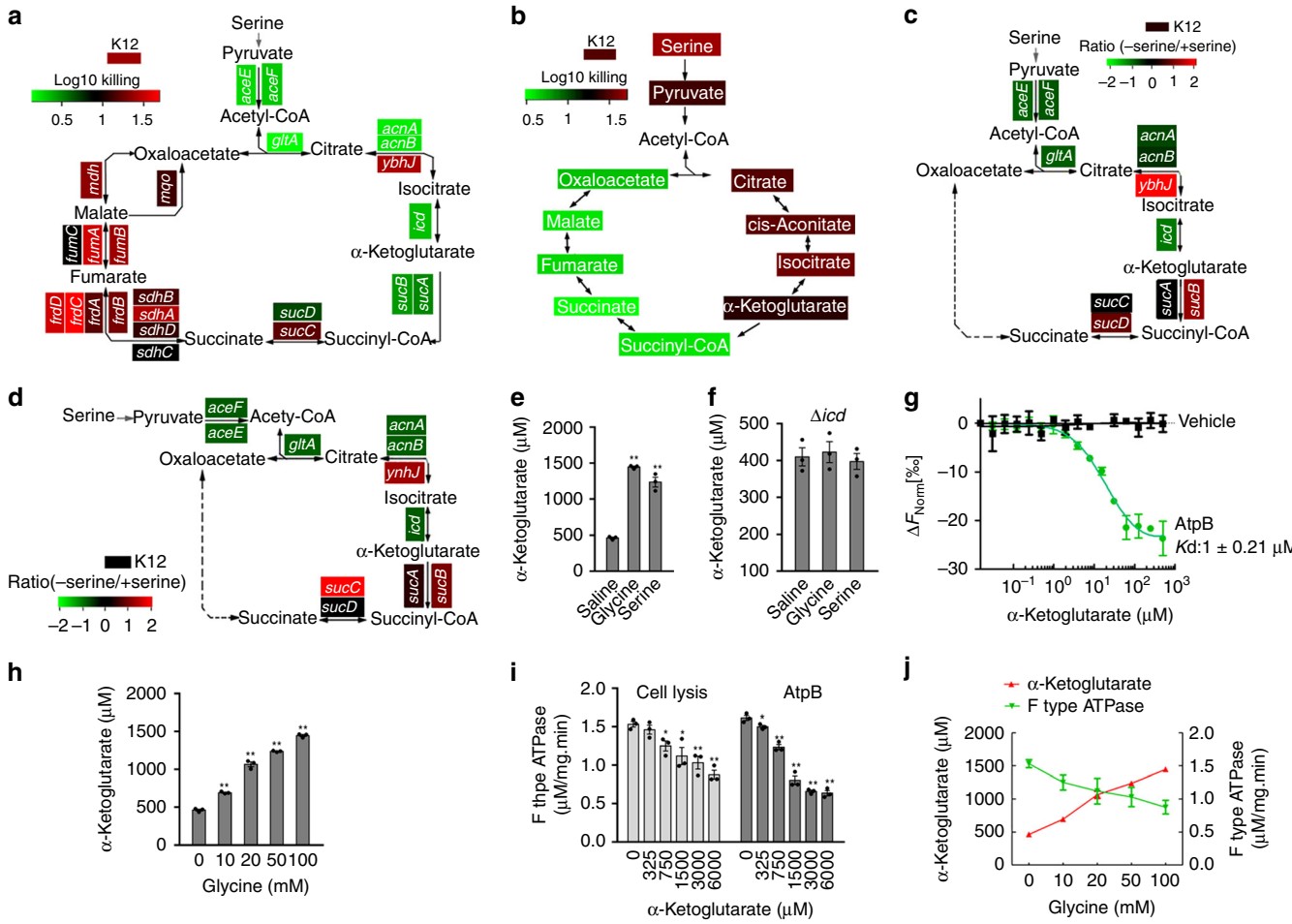

**Fig. 5** The TCA cycle regulates bacterial serum susceptibility. **a** Summary of relative survival of the indicated mutant strains of *E. coli* K12 in the presence or absence of 50 mM serine plus 100 μL serum for 2 h (*n* = 3). **b** Summary of relative survival of *E. coli* K12 in the presence or absence of the indicated metabolites of the TCA cycle plus 100 μL serum for 2 h (*n* = 3). **c, d** Summary of relative ATP synthase activity (**c**) and ATP content (**d**) of the indicated mutant strains of *E. coli* K12 for 2 h in the presence and absence of 50 mM serine plus 100 μL serum. Heat map scale (**a**–**d**) (green to red) represents low to high killing (**a**, **b**) or ratio (**c**, **d**) without the indicated genes (**a**, **c** and **d**) or with the indicated metabolites (**b**) (*n* = 3). **e**, **f** Intracellular α-Ketoglutarate concentration in the presence or absence of 100 mM glycine or serine (**e**), and the effect of loss of *icd* (**f**) (*n* = 3). **g** AtpB and α-Ketoglutarate interaction assay by MST (*n* = 3). **h** Effect of exogenous glycine (0–100 mM) on the intracellular α-Ketoglutarate level (*n* = 3). **i** Measurement of intracellular ATP synthase activity using *E. coli* K12 cell lysis (cell lysis) or purified recombinant AtpB plus the indicated concentration of α-Ketoglutarate (*n* = 3). **j** The interrelation between α-Ketoglutarate level and intracellular ATP synthase activity from data (**h**, **i**) (*n* = 3). Results (**e**–**j**) are displayed as mean ± SEM, and significant differences are identified (*$p < 0.05$, **$p < 0.01$) as determined by two-tailed Student's *t* test (**e**, **h**, **i**). See also Supplementary Fig. 9

cell viability (Fig. 4g). These results together support that glycine catabolism to the TCA cycle is critical to reverse serum resistance.

**α-Ketoglutarate regulates ATP synthase.** Following serine flux to the TCA cycle via pyruvate, we found gene-specific effects. In particular, the presence of genes in the pyruvate metabolism (*aceE* and *aceF*) and in the upper TCA cycle (*gltA*, *acnA*, *acnB*, *icd*, *sucA*, and *sucB*) induced rapid killing of bacteria by serum. In contrast, the presence of genes in the lower TCA cycle lacked this effect (Fig. 5a; Supplementary Fig. 9). These results were supported by the observation of increased serum-sensitivity in the presence of exogenous pyruvate, citrate, cis-aconitate, isocitrate or α-ketoglutarate, but not succinyl-CoA, succinate, fumarate, malate or oxaloacetate (Fig. 5b). To explain these results, we postulate that a defective step in the TCA cycle might lead to a defect in ATP synthesis. To explore this, ATP synthase activity was measured in the presence or absence of serine in *E. coli* K12, Δ*aceE*, Δ*aceF*, Δ*gltA*, Δ*acnA*, Δ*acnB*, and Δ*icd*; these results show a significant effect in these mutants, but not other single deletion

mutants tested here. The final gene in the pathway, *icd*, converts isocitrate to α-Ketoglutarate (Fig. 5c). Consistent with this, the amount of ATP was decreased in these strains (Fig. 5d). Glycine and serine increased α-Ketoglutarate concentration (Fig. 5e) but not in Δ*icd* cells (Fig. 5f). These results indicate that α-Ketoglutarate might inhibit ATP synthase. To further demonstrate this, microscale thermophoretic analysis (MST) showed that α-Ketoglutarate interacts with AtpB, a subunit of ATP synthase (Fig. 5g). Exogenous glycine elevated intracellular α-Ketoglutarate in a dose-dependent manner, from 480 μM, a concentration reported previously[39], to 750, 110, 1300, and 1450 μM by 10, 20, 50, and 100 mM glycine, respectively (Fig. 5h), leading to 270–970 μM (1.56–3.20 folds) of α-Ketoglutarate increase. The inhibition of α-Ketoglutarate to ATP synthase appeared in 750 μM and 325 μM α-Ketoglutarate in cell lysis and purified AtpB, respectively (Fig. 5i). Clearly, exogenous glycine promoted the abundance of α-Ketoglutarate but inhibited ATP synthase activity in a dose-dependent manner (Fig. 5j). Thus, this accumulation induced by exogenous glycine is enough to inhibit ATP synthase.

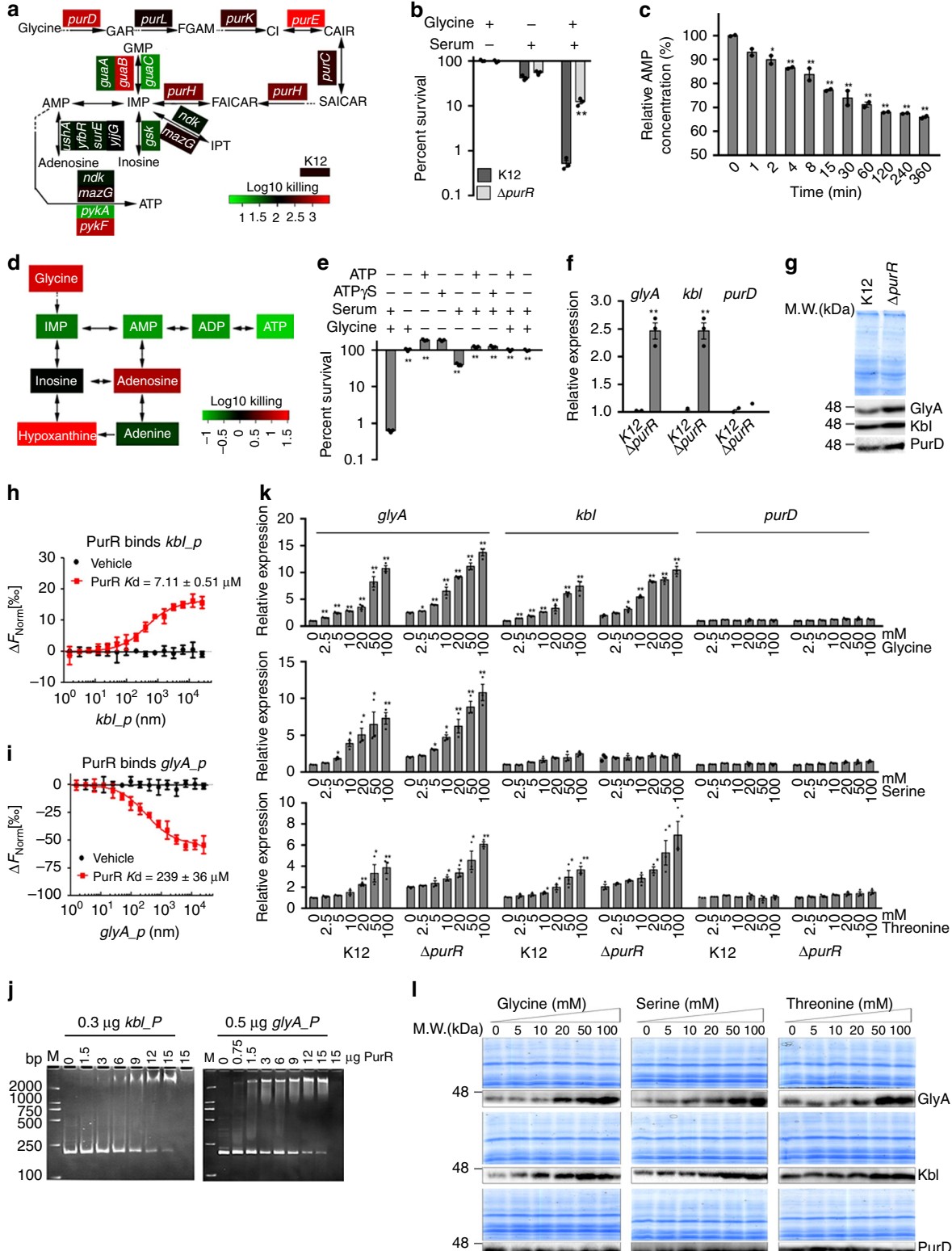

## Exogenous glycine regulates purine metabolism.

On the other hand, besides ATP synthase, ATP synthesis needs AMP and ADP. Therefore, a role for purine metabolism was explored, because mutations in *purD*, a gene involved in the de novo purine nucleotide biosynthesis, altered serum resistance (Fig. 4a), and reports have linked glycine metabolism with de novo purine synthesis[40,41]. A series of mutant strains of *E. coli* K12, each deleted for a gene in the purine biosynthetic pathway, were

cultured in the presence and absence of glycine plus serum. The results show decreased viability of most strains with defects in purine biosynthesis (Fig. 6a), suggesting that purine biosynthesis influences serum-dependent cell death. We validated this by demonstrating that Δ*purR* cells, lacking a potent negative regulator of purine nucleotide metabolism, are more resistant to cell killing in the presence of serum and glycine than *E. coli* K12 (Fig. 6b; Supplementary Fig. 9). We supposed that this led to

**Fig. 6** Regulation of glycine, serine, and threonine to GlyA. **a** Percent survival of *E. coli* mutants lacking the indicated genes of purine metabolism in the presence or absence of 100 mM glycine plus 100 μL serum. Heat map scale (green to red) represents low to high killing without the indicated genes ($n = 3$). **b** Percent survival of Δ *purR* in the presence of 100 mM glycine, 100 μL serum or both ($n = 3$). **c** Relative AMP abundance of *E. coli* K12 in the presence of 100 mM glycine for the indicated length of time ($n = 2$). **d** Percent survival of *E. coli* K12 in the presence of the indicated purine metabolites (100 mM glycine, 2 mM IMP, 4 mM AMP, 2.5 mM ADP, 5 mM ATP, 10 mM inosine, 10 mM adenosine, 10 mM hypoxanthine, 10 mM adenine) plus 100 μL serum. Heat map scale (green to red) represents low to high killing with the indicated metabolites ($n = 3$). **e** Percent survival of *E. coli* K12 in the presence or absence of the indicated metabolites with or without serum. ATPγS, a nonhydrolyzable ATP analog ($n = 3$). **f** qRT-PCR for relative expression of *glyA*, *kbl*, and *purD* in the presence or absence of *purR* ($n = 3$). **g** Western blot for GlyA, Kbl and PurD expression in the presence or absence of *purR*. **h, i** MST assay for investigation of PurR-*kbl* promoter (**h**) and PurR-*glyA* promoter interaction (**i**) ($n = 3$). **j** Electrophoretic mobility shift assay (EMSA) for investigation of PurR-*kbl* promoter and PurR-*glyA* promoter interaction. **k** qRT-PCR for relative expression of *glyA*, *kbl*, and *purD* in the presence or absence of *purR* and in the indicated concentrations of glycine, serine or threonine ($n = 3$). **l** Western blot for GlyA, Kbl and PurD expression in the indicated concentrations of glycine, serine, or threonine. Results in (**b, c, e, f, k**) are displayed as mean ± SEM, and significant differences are identified (*$p < 0.05$, **$p < 0.01$) as determined by two-tailed Student's *t* test (**b, c, e, f, k**). See also Supplementary Fig. 9

decreasing AMP level and thereby exogenous AMP and ATP might abolish the role of glycine. Consistent with the hypothesis, we detected a glycine-dependent decrease of AMP with time (Fig. 6c), and demonstrated that exogenous IMP, AMP, ADP, and ATP abolish glycine-dependent bacterial cell death (Fig. 6d). ATP analog had a similar effect as ATP (Fig. 6e). This finding suggests that ATP contributes to serum resistance.

PurR also negatively regulates *glyA*[42]. We also show that loss of *purR* led to increased expression of *glyA*/GlyA and *kbl*/Kbl using qRT-PCR and Western blot (Fig. 6f, g). MST and electrophoretic mobility shift assay (EMSA) confirmed the binding of PurR with *glyA* and *kbl* (Fig. 6h–j). However, the results on the negative regulation do not explain why exogenous glycine, serine, and threonine flux to the TCA cycle rather than purine metabolism via PurD. One possibility is that these three metabolites positively regulate GlyA or/and Kbl but not PurD through substrate activation, which is over the action negatively regulated by PurR. To explore this, qRT-PCR was used to detect expression to *glyA*, *kbl*, and *purD* in *E. coli* K12 and Δ*purR* cultured in medium with different concentrations of glycine, serine, or threonine. High dose of these metabolites favored increasing *glyA* and *kbl* expression in *E. coli* K12. In details, the expression of *glyA* and *kbl* were elevated by glycine and threonine, and the expression of *glyA* was increased by serine, whereas no significant change was detected in the expression of *purD* in response to three metabolites (Fig. 6k). Consistent with the negative regulation of PurR to *glyA* and *kbl*, lower *glyA* and *kbl* expression was detected in *E. coli* K12 than Δ*purR* (Fig. 6k). Western blot data also validated these findings, showing GlyA was increased in a dose-dependent manner, while the weak elevation of Kbl and no significant change of PurD were detected (Fig. 6l), and *purD*/PurD was not regulated by PurR (Fig. 6f, g). These results indicate that high dose of exogenous glycine leads to increased intracellular serine but lower purine biosynthesis; this may reflect the expression of elevated GlyA and unchanged PurD.

**Mechanism of cell death in the presence of serum and glycine**. These above results suggest that ATP plays a role in the serum-mediated killing. To demonstrate this, we detected a glycine-dependent decrease of ATP over time lapse (Fig. 7a) and found that exogenous ATP abolished glycine-dependent bacterial cell death (Fig. 7b). We also found that intracellular ATP was positively regulated by exogenous ATP, ATPγS, ADP, AMP, but negatively regulated by serum, glycine/serum (Supplementary Fig. 10b).

ATP is converted into cyclic-AMP (cAMP) by adenylate cyclase (coded by *cyaA*)[43]. cAMP and CRP (coded by *crp*) form cAMP-CRP complex, a global transcriptional regulator controlling a minimum of 378 target promoters in *E. coli*[44]. Consistent with this, synergistic action of glycine and serum led to a decreased abundance of cAMP, whereas single glycine or serum did not affect cAMP level (Fig. 7c); loss of *cyaA* and *crp* led to decreased viability (Fig. 7d; Supplementary Fig. 9). We also demonstrated that the abundance of CRP increases in an ATP- and AMP-dependent manner (Fig. 7e), whereas CRP expression is downregulated by glycine (Fig. 7f). These results suggest that exogenous glycine may decrease the abundance of the cAMP-CRP complex in cells exposed to serum, which is supported by the fact that absence of *fis* increases viability in the presence of glycine since *fis* negatively regulates *crp* (Fig. 7d; Supplementary Fig. 9)[45].

CRP positively and negatively regulates *flhC* and *csgD*, two genes that regulate fimbrial and pilus usher proteins, respectively[46,47]. HtrE and YhcD are fimbrial usher proteins and NfrA is a bacteriophage N4 receptor with similar molecular mass as usher proteins. CRP regulates expression of outer membrane proteins in the manner of FlhC and CsgD[47]. Consistent with this, loss of *flhC* and *csgD* led to upregulation and downregulation of NfrA, HtrE, and YhcD, respectively (Fig. 7g) and the respective percent survival (Fig. 7d). Moreover, the expression of *nfrA*/NfrA, *htrE*/HtrE, and *yhcD*/YhcD was higher in a *crp*-deleted mutant than in wild type control (Fig. 7g, h). The results suggest that CRP regulates expression of NfrA, HtrE, and YhcD in the manner of FlhC and CsgD. In addition, loss of *ompA* and *ompC* led to lower percent survival in the presence of serum but not in the presence of serum and glycine (Supplementary Fig. 11).

Flow cytometry analysis confirmed significantly lowest binding of anti-C9 neoantigen of the terminal complement complex (TCC, MAC, the whole C5b-9 complex) to the triple mutants, Δ*nfrA*Δ*yhcD*Δ*htrE*, Δ*htrE*Δ*yhcD*Δ*nfrA*, and Δ*yhcD*Δ*htrE*Δ*nfrA*, lower binding to the double mutants Δ*nfrA*Δ*yhcD*, Δ*htrE*Δ*yhcD*, Δ*yhcD*Δ*htrE*, and low binding to the single mutants Δ*nfrA*, Δ*htrE*, and Δ*yhcD* as compared to control (Fig. 7i). Corresponding percent survival was detected in these mutants (Fig. 7j). These are consistent with the hypothesis that HtrE, NfrA, and YhcD could be the outer membrane proteins that promote binding of complement components in the presence of exogenous glycine. Western blot analysis was then performed to measure relative expression of HtrE, NfrA, and YhcD in *E. coli* K12 grown in PBS, serum or in serum plus glycine. In the presence of both of serum and glycine, expression of HtrE, NfrA, and YhcD increased in a serum and glycine dose-dependent manner; in contrast, expression of HtrE, NfrA, and YhcD remained unchanged in the presence of media with added glycine, serum or saline (Fig. 7k, l). Similar results were obtained with *E. coli* Y17 and four pathogenic *E. coli* strains (Fig. 7m; Supplementary Fig. 12). These results indicate that glycine in the presence of serum stimulates expression of HtrE, NfrA, and YhcD.

To investigate whether the increased expression of HtrE, NfrA, and YhcD is contributing to the increased binding of components

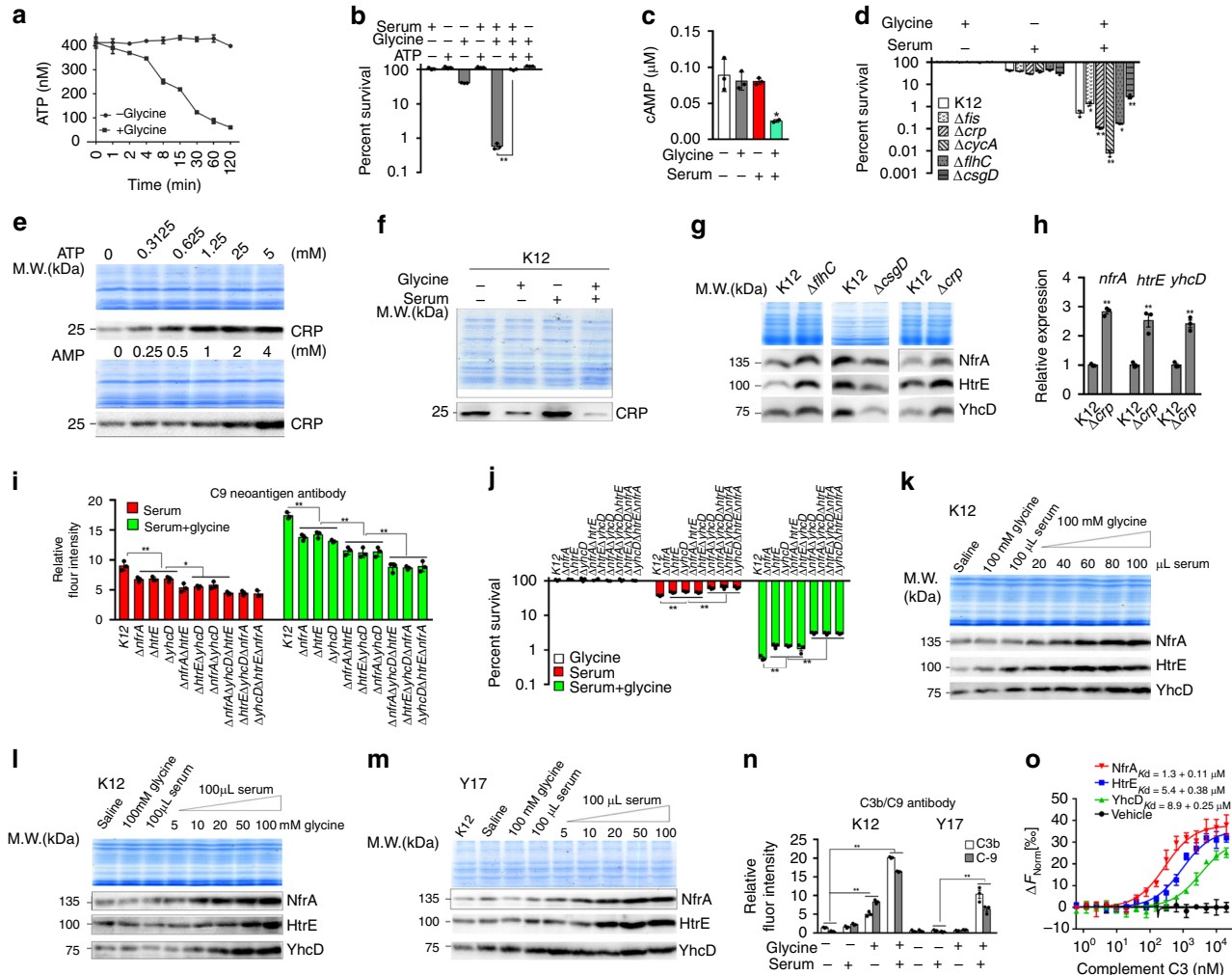

**Fig. 7** Glycine-dependent mechanism for HtrE, NfrA, YhcD expression. **a** Relative ATP abundance of *E. coli* K12 in the presence or absence of 100 mM glycine for the indicated length of time ($n = 3$). **b** Percent survival of *E. coli* K12 in the presence or absence of 5 mM ATP, 100 mM glycine or/and 100 μL serum ($n = 3$). **c** Relative cAMP abundance of *E. coli* K12 in the presence or absence of 100 mM glycine or/and 100 μL serum ($n = 3$). **d** Percent survival of the indicated mutants in the presence of 100 mM glycine, 100 μL serum or both ($n = 3$). **e** Western blot of CRP in *E. coli* K12 exposed to 0–5 mM ATP or 0–4 mM AMP. **f** Effect of 100 mM glycine, 100 μL serum or both on CRP expression. **g** Western blot of HtrE, NfrA, YhcD in these mutants. **h** qRT-PCR of *htrE*, *nfrA*, *yhcD* in these mutants ($n = 3$). **i** Flow cytometry quantification of anti-C9 neoantigen on outer membrane surface of the indicated mutants ($n = 3$). **j** Percent survival of the indicated mutants in the presence of 100 mM glycine, 100 μL serum or both ($n = 3$). **k** Western blot of HtrE, NfrA, YhcD of *E. coli* K12 in the presence of 100 mM glycine and the indicated amount of serum. **l, m** Western blot of HtrE, NfrA, YhcD of *E. coli* K12 (**l**) and Y17 (**m**) in the presence of 100 μL serum and the indicated concentrations of glycine. **n** Flow cytometry quantification of anti-C3b and anti-C9 neoantigen on *E. coli* K12 and Y17 in the presence or absence of 100 μL serum or 100 mM glycine or both. **o**, MST for the interaction of complement C3 with HtrE, NfrA, YhcD ($n = 3$). **i, n** 50,000 cells were record with forwarding versus side scatter and were gated before data acquisition. Results in (**a–d, h–j, n, o**) are displayed as mean ± SEM, and significant differences are identified (*$p < 0.05$, **$p < 0.01$) as determined by two-tailed Student's *t* test. See also Supplementary Figs. 10–15

of the complement system, flow cytometry was used. Serum or serum plus glycine led to the deposition of C3b and C5b-9 complex on the outer membrane of *E. coli* K12 and Y17 (Fig. 7n). In particular, anti-C3b/iC3b and anti-C9 neoantigen recognized intact C3b, its degraded product, iC3b, and a neoepitope of TCC on the surface of serum or serum plus glycine-exposed *E. coli* K12 and Y17, respectively. Furthermore, TCC was detected at higher abundance by flow cytometry, when cell killing was more efficient, such that more C3b/TCC was present on the surface of *E. coli* K12 and Y17 exposed to serum plus glycine than on the surface of control cells exposed to saline, glycine, or serum. Consistent with this observation, more C3b/TCC was present on the surface of *E. coli* K12 than on Y17 after exposure to serum or serum plus glycine (Fig. 7n). MST assay demonstrated that HtrE,

NfrA, and YhcD were bound to C3 (Fig. 7o). Far-Western blot and co-immunoprecipitation also showed that HtrE, NfrA, and YhcD interact with C3b, but not with IgG control (Supplementary Fig. 13). Therefore, glycine potentiates killing of serum-resistant bacteria, possibly by upregulating HtrE, NfrA, and YhcD, enabling the complement-dependent MAC-mediated cell death.

In addition, we investigated effect of O-antigen on the glycine-mediated killing, which is critical for LPS-mediated serum resistance in *E. coli*[10,19,20]. Among the 30 clinical *E. coli* strains, 27 strains were O-antigen positive (Supplementary Table 3). Waal works for LPS biosynthesis and is subjected to frequent mutations to generate serum resistance[47]. The mutations of *waal* were detected in randomly selected *E. coli*, *P. aeruginosa*, and

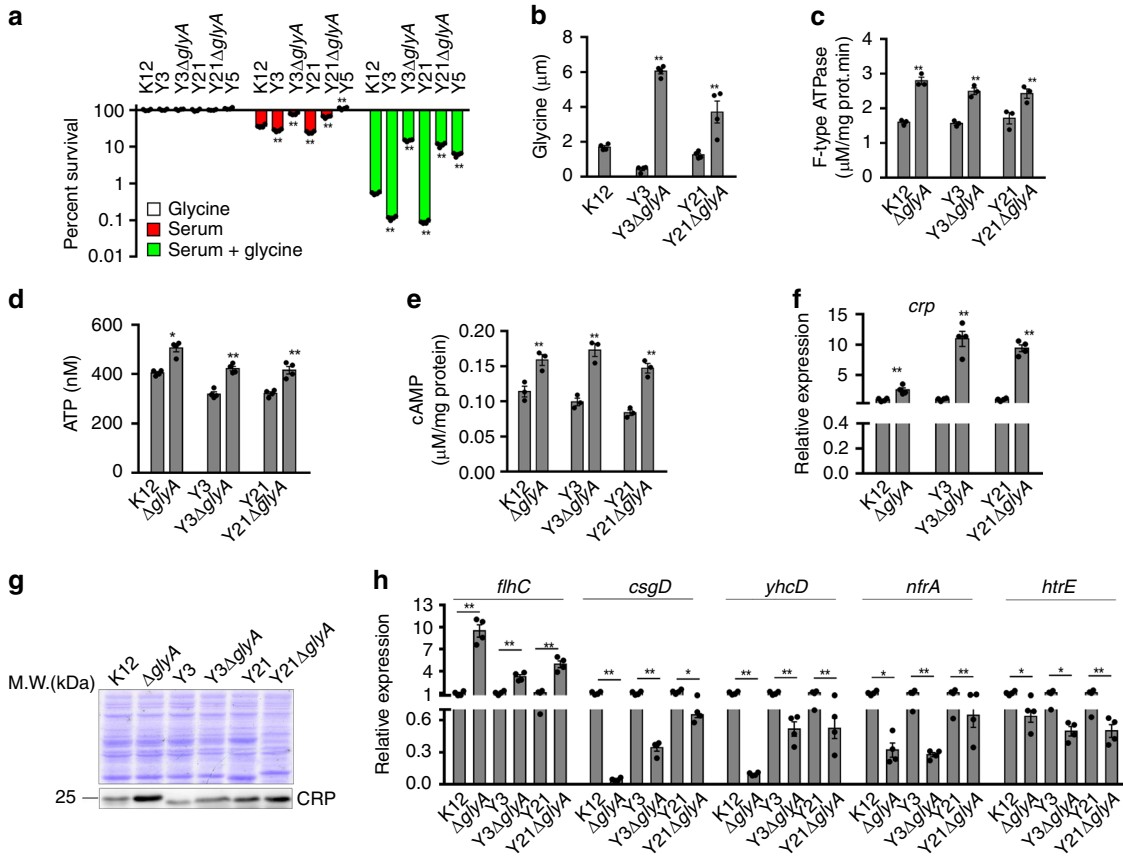

**Fig. 8** Effect of *glyA* deletion on glycine-cAMP/CRP-YhcD/NfrA/HtrE pathway. **a, b** Percent survival (**a**) and intracellular glycine (**b**) of *E. coli* Y3 and Y21 and their *glyA*-deleted mutants Y3△*glyA* and Y21△*glyA*, respectively. *E. coli* K12 and serum-resistant Y15 were used as negative and positive controls, respectively ($n = 4$). **c–e** ATP synthase activity (**c**), ATP level (**d**), and cAMP level (**e**) in the indicated strains (**c, e,** $n = 3$; **d,** $n = 4$). **f** *crp* expression in the absence of *glyA* ($n = 4$). **g** CRP expression in the absence of *glyA*. **h** qRT-PCR for expression of *flhC, csgD, yhcD, nfrA,* and *htrE* in the absence of *glyA* ($n = 3$). Results are displayed as mean ± SEM (**a–f, g**), and significant differences are identified (*$p < 0.05$, **$p < 0.01$) as determined by two-tailed Student's *t* test (**a–f, g**)

*K. pneumonia* strains (Supplementary Fig. 14). *E. coli* serotype O157:H7, with adhesive fimbriae and a cell wall that consists of an outer membrane containing LPS, was sensitive to the glycine-triggering killing (Supplementary Fig. 15a) while exogenous glycine did not affect *waal* transcription (Supplementary Fig. 15b). When *waal* was deleted, the glycine-mediated killing was either unaffected or reduced by 4.81–24.1% (Supplementary Fig. 15c). These results indicate that LPS is not the key target in the glycine-triggering serum killing.

**Loss of *glyA* confers serum resistance in serum-sensitive strains.** To validate this possibility at a physiological level, we constructed *glyA* single deletion mutants in two serum-sensitive strains Y3 and Y21 as Y3Δ*glyA* and Y21Δ*glyA*, respectively. Y3 and Y21 were more susceptible to serum than *E. coli* K12, but Y3Δ*glyA* and Y21Δ*glyA* were highly resistant to serum at a similar level of a serum-resistant strain Y15. When exogenous glycine was supplemented, percent survival was reduced in all the five strains. However, higher killing efficacy was detected in Y21, Y3, K12 than Y15, Y21Δ*glyA*, Y3Δ*glyA* (Fig. 8a). Correspondingly, *glyA* deletion resulted in increased glycine level in both Y3Δ*glyA* and Y21Δ*glyA* (Fig. 8b). However, the increased glycine level did not inhibit ATPase activity and ATP production (Fig. 8c, d) and thereby increased the cAMP level (Fig. 8e). These results indicate that serum-susceptible bacteria exhibit serum resistance when no glycine fluxes to the TCA cycle.

CRP is a transcriptional factor modulated by cAMP. We further explored whether loss of *glyA* would impact CRP activity. qRT-PCR and Western blot showed that the deletion increased CRP expression at both mRNA and protein level (Fig. 8f, g). CRP positively and negatively regulates FlhC and CsgD, respectively, which downregulate the expression of *nfrA*, *htrE*, and *yhcD*[46,48,49]. Consistently, the deletion of *glyA* resulted in increased expression of *flhC* but decreased expression of *csgD*, *nfrA*, *htrE*, and *yhcD* (Fig. 8h). These results indicate that the elevated cAMP/CRP when *glyA* was depleted confers serum resistance through reducing NfrA, HtrE, and YhcD expression. Therefore, the metabolic flow of glycine to the TCA cycle is an essential step for sensitivity to serum-mediated killing.

**The potential is effective to serum-resistant pathogens.** To explore whether the glycine-mediated serum killing was applicable to a broad range of pathogens, more studies were performed on clinically-relevant multidrug-resistant *E. coli*, *Klebsiella pneumonia*, methicillin-resistant *Staphylococcus aureus* (MRSA), and *Pseudomonas aeruginosa*. These pathogenic bacteria were cultured in vitro in the absence or presence of serum or serum plus glycine. With the exception of a few *E. coli* strains and two strains of *P. aeruginosa*, all strains were serum-resistant. Cell death was 10- to 1430-fold greater in the presence of glycine and serum than in the presence of serum alone. The average (and range) of this effect was 143.97-fold (24.65–936.43) in *E. coli*, 67.08-fold (15.17–122.19) in *K. pneumoniae*, 49.28-fold (10.0–95.6) in

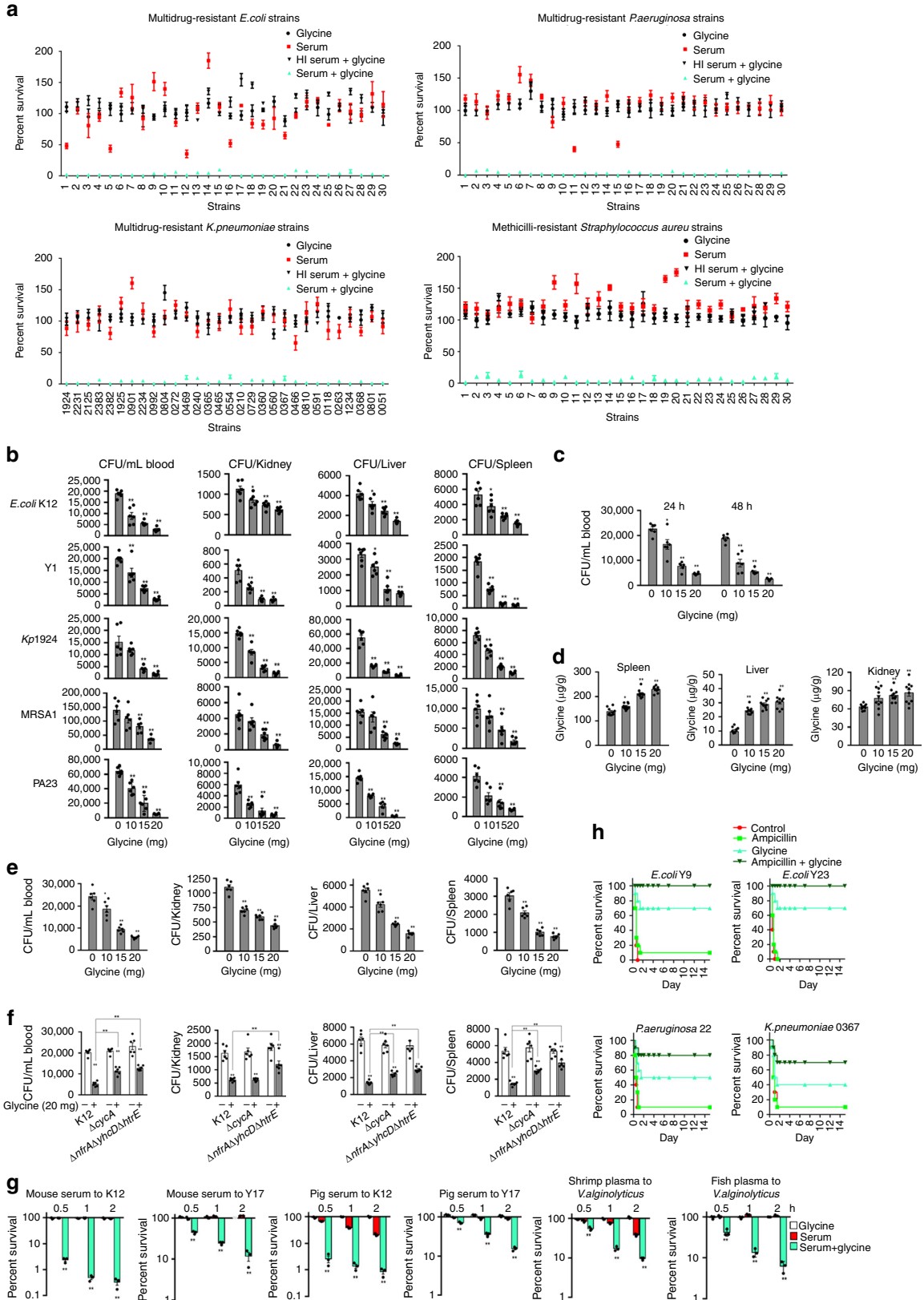

MRSA, and 230.41-fold (11.76–1430.83) in *P. aeruginosa* (Fig. 9a). Notably, MRSA is a Gram-positive bacterium deficient of outer membrane targeted by the MAC complex. Our results showed that higher level of C3 and C5 on the surface when glycine was supplemented (Supplementary Fig. 16), which requires further study.

A mouse model was used to explore the efficacy of glycine for inhibiting pathogenic infections in vivo. Glycine was delivered by intravenous injection at a dose of 10, 15, or 20 mg glycine/mouse twice a day for 2 days. BALB/c mice were infected with *E. coli* strains K12, Y1, and Y17, *K. pneumoniae* strains 1924 and 2125, MRSA strains 1 and 5, and *P. aeruginosa* strains 1 and 23

**Fig. 9** Glycine increases the susceptibility of clinical bacteria to killing by serum. **a** Percent survival of multidrug-resistant *E. coli*, *P. aeruginosa*, *K. pneumoniae*, and MRSA in the presence of 100 mM glycine, 100 μL serum, or both, or 100 μL HI serum plus 100 mM glycine. Differences were found at $p < 0.01$ between the group with glycine + serum and the three other groups in all samples ($n = 3$). **b**, **c** BALB/c mice were infected with the indicated bacteria by *i.p.* injection (see methods) and treated with glycine as described (see text). Bacterial load was measured in blood, kidney, spleen, and liver (**b**) and with time in the blood (**c**) ($n = 6$). **d** Glycine concentrations in the spleen, liver, and kidney were measured at 48 h after the *i.p.* injection of the indicated glycine concentration ($n = 10$). **e**, **f** $Rag1^{-/-}$ (**e**) and BALB/c (**f**) mice were infected with the indicated bacteria by *i.p.* injection (see methods) and treated with glycine as described (see text). Bacterial load was measured in blood, kidney, liver, and spleen ($n = 6$). **g** Glycine-enabled killing of the indicated bacteria by 100 μL other sources of serum with or without 100 mM glycine for the indicated length of time ($n = 3$). **h** Percent survival of mice infected the indicated bacterial pathogens and then treatment by saline (control), ampicillin (320 mg kg$^{-1}$), glycine (800 mg kg$^{-1}$) or both as described (see text for details) ($n = 6$). Results are displayed as mean ± SEM, and significant differences are identified (*$p < 0.05$, **$p < 0.01$) as determined by two-tailed Student's *t* test. See also Supplementary Figs. 19–20

via intraperitoneal injection. The results showed that bacterial load in the bloodstream, kidney, spleen, and liver of infected mice decreases in a glycine dose- and time-dependent manner (Fig. 9b, c, Supplementary Fig. 17). In the plasma, the concentration of glycine was increased dramatically and reached the plateau within 5 min after the first injection. Then the level was gradually decreased and was returned to the baseline after 1 h post-injection. A second injection was performed at 12 h after the first injection. A similar glycine change level was observed but with a bit of higher abundance (Supplementary Fig. 18). In contrast, the level of glycine was elevated in liver, kidney, and spleen in a dose-dependent manner at 48 h after the first injection (Fig. 9d). The glycine-enabled killing of bacteria by serum was detected in $Rag1^{-/-}$ null mice which do not bear mature T cells or B cells, supporting the glycine-mediated potential is independent of immunity endowed by the mature T cells or B cells (Fig. 9e). To demonstrate the role of glycine and complement in the potentiation, BALB/c mice were injected with K12, ΔcycA, or ΔnfrAΔyhcDΔhtrE, respectively. Bacterial load in the bloodstream, spleen, and liver was higher in ΔcycA-infected or ΔnfrAΔyhcDΔhtrE than K12-infected mice (Fig. 9f), thereby validating that glycine, and HtrE, NfrA, YhcD play roles.

We also showed that *E. coli* K12 cells remained inviable in vitro in the presence of serum plus glycine after five serum-killing/dilution-culture cycles, while showing weak susceptibility to serum in the absence of glycine (Supplementary Fig. 19).

Survival of *E. coli* K12 and Y17 decreased significantly in media with glycine plus serum from mouse and pig, as did survival of *V. alginolyticus* in media with glycine plus fish and shrimp plasma (Fig. 9g). This result suggests that the metabolic pathways and the biological basis of serum-dependent cell death in the presence of exogenous glycine are similar in mammals and some non-mammalian species.

We further explored whether glycine potentiates antibiotics to eliminate serum-susceptible or serum-resistant and multidrug-resistant bacterial pathogens. Percent survival of *E. coli* was lower in glycine with ampicillin or kanamycin than only glycine or the drugs (Supplementary Fig. 20). Mice were infected with pathogenic *E. coli* Y9 (serum-resistant and multidrug-resistant), *E. coli* Y23 (serum-susceptible and multidrug-resistant), *P. aeruginosa* 22 (serum-resistant and multidrug-resistant) or *K. pneumonia* 0367 (serum-resistant and multidrug-resistant). The synergistic use of glycine and ampicillin increases the survival rate of the infected mice. Specifically, mice infected with *E. coli* Y9 or *E. coli* Y23 had a 100% survival rate if both glycine and ampicillin were treated, and a 70% survival rate if the only glycine was treated, while the survival rate dropped to 0–10 % if treated with ampicillin alone or with saline. Similarly, mice infected with *P. aeruginosa* 22 or *K. pneumonia* 0367 had a survival rate of 70–80% when both of ampicillin and glycine were treated, and a 40–50% survival rate if only glycine was treated, while 10% survival was found if treated with ampicillin alone or

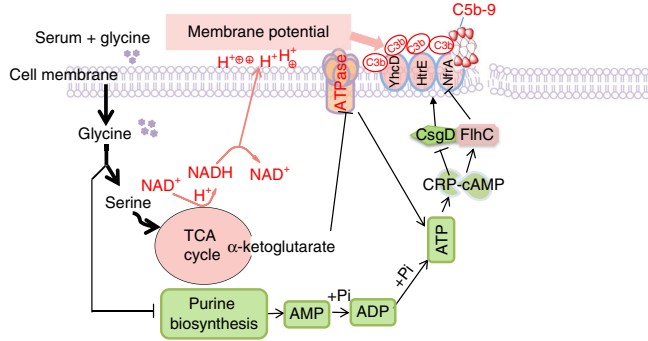

**Fig. 10** Proposed mechanism of glycine-promoted complement-dependent killing. First, exogenous glycine passes into the cytoplasm, where it is converted to serine and threonine by GlyA and Kbl, respectively; second, serine fluxes to the TCA cycle and promotes α-Ketoglutarate generation, which inhibits ATP synthase, while elevated glycine, serine and threonine catabolic pathway does not favor fluxing purine metabolism due to stronger substrate activation to GlyA, thereby decreasing AMP and ADP biosynthesis. These lead to ATP reduction and the decrease of ATP affects cAMP and cAMP-CRP complex; third, glycine in conjunction with serum elevates expression and membrane deposition of HtrE, NfrA, and YhcD by FlhC and CsgD in a manner of cAMP-CRP complex; and fourth, these outer membrane proteins promote binding of complement, formation of MAC and subsequent cell death. On the catabolic side, glycine enters the TCA cycle and stimulates synthesis of NADH and increases membrane potential, which facilitates the binding of complement to the outer membrane. It is clear that metabolites dominate the novel demonstrated metabolic regulation pathways. Colors: pink, upregulation; grass green, down-regulation

with saline (Fig. 9h). These results indicate that glycine promotes ampicillin to kill serum-susceptible or serum-resistant and multidrug-resistant pathogens.

## Discussion

Serum resistance as bacterial pathogenesis was found more than a hundred years, but information regarding effective controls is not available. The present study highlights the way to eliminate serum-resistant bacteria through metabolome-reprogramming. The core findings of this study are: (1) downregulation of glycine, serine and threonine catabolism to the TCA cycle is a characteristic feature of serum-resistant bacteria; (2) glycine, serine and threonine catabolism, the TCA cycle, purine metabolism, ATP synthase, and ATP abundance, as well as PMF, play roles in serum-mediated cell death; (3) the cAMP/CRP complex modulates expression of HtrE, NfrA, and YhcD, which promotes complement binding, leading to MAC-mediated bacterial cell death; (4) the latter pathway plays a key role in serum-mediated cell death. These findings are summarized in (Fig. 10).

In the present study, two novel findings are interesting. First, a novel metabolic regulation pathway is disclosed. Glycine is consecutively transformed to α-Ketoglutarate and then α-Ketoglutarate interacts with ATP synthase. It is an interesting finding that bacterial α-Ketoglutarate inhibits ATP synthase, demonstrating a mechanism to maintain membrane potential that involves metabolite feedback. Second, we suggest that metabolite feedback may override the effects of global gene regulation. More specifically, although PurR negatively regulates *glyA* and *kbl*, a high concentration of glycine, serine, or/and threonine increased expression of *glyA*/GlyA and *kbl*/Kbl several-fold. In addition, the regulation of PurR to *kbl* and *glyA* to *crp*/CRP is found. And HtrE, NfrA, and YhcD are unknown complement-binding outer membrane proteins.

## Methods

**Ethics statement**. Animal protocols were approved by the University Committee for the Use and Care of Laboratory Animals. All work was conducted in strict accordance with the recommendations of the Guide for the Care and Use of Laboratory Animals of the National Institutes of Health.

**Bacterial strains and growth conditions**. In this study, the parent strain *E. coli* K12 BW25113 (genotype, Δ(*araD-araB*)567, Δ*lacZ4787*(::*rrnB*-3), lambda⁻, *rph-1*, Δ(*rhaD-rhaB*)568, *hsdR*514) and its knockout strains were taken from the KEIO collection. A single colony of *E. coli* K12 BW25113 and its knockout strains were picked from the Luria-Bertani (LB) agar plate. Reliability of these mutants was validated by gene sequencing, which primers were listed in Supplementary Table 4. These bacteria were separately inoculated in 5 mL of LB medium for 16 h at 37 °C and spun at 200 rpm. Overnight cultures were diluted 1:100 in 50 mL fresh LB medium and grew to phase OD₆₀₀ of 0.5 at 37 °C. Bacterial cells were collected by centrifugation at 8000 × *g* for 10 min at 4 °C and suspended in saline solution. Unless otherwise noted, 100 mM glycine was added and cultured at 200 rpm for 2 h in 37 °C in all experiments. Samples were pre-treated with the proton ionophore CCCP for 5 min in all experiments. The other bacteria, multidrug-resistant *E. coli*, *P. aeruginosa*, *K. pneumonia*, methicillin-resistant *Staphylococcus aureus* (MRSA), *V. alginolyticus* and *V. parahaemolyticus* were from the collections of our laboratory. These cells were insensitive to at least three classes of antibiotics including β-lactamases, aminoglycoside, quinolones or/and sulfonamides. Unless otherwise noted, the culture conditions were the same above. All bacterial strains used were listed in Supplementary Table 5.

**Serum killing**. Human serum was collected from 200 healthy donors. The collected sera were packaged and kept at −80 °C for future use. Package sera were not used repeatedly when they were taken from the refrigerator. The collected bacteria above were suspended with sterile saline and adjusted to OD₆₀₀ of 1.0. Aliquots 3 mL of the bacteria were added in a 5 mL centrifugation tube and were centrifuged at 8,000 g for 10 min in 4 °C. The remnants of normal saline were removed by pipette. Then, 100 μL serum, C3-depleted serum (purified by anti-human C3 affinity chromatography), HI serum or serum treated with 10 mM EDTA or 20 mM EGTA drying in 5 mL tube and 150 μL saline solution containing desired metabolites were added as an experiment group and an equal concentration of saline solution was used for a control group. The mixtures were cultured at 200 rpm at 37 °C for 2 h. The residual cells were collected by centrifugation at 8,000 rpm for 5 min in 4 °C and were then re-suspended in 3 mL saline solution. The samples were serially diluted with sterile saline, and 10 μL aliquots were spot-plated onto LB agar plates. The plates were cultured at 37 °C and the colony-forming unit (CFU) were calculated the next day. Only those of dilutions yielding 20–200 colonies were enumerated to calculate CFU. The percent survival was determined by dividing CFU of the treatment sample by CFU of control. Experiments were repeated in at least three independent biological replicates.

**Metabolomic profiling**. GC-MS analysis was carried out with a variation on the two-stage technique[25]. Peak intensities were normalized to form a single matrix with Rt-*m/z* pairs for each file in the dataset. Statistical analysis was performed as follows[50,51]. The data were median centered and inter-quartile range (IQR) scaled per sample. Plotted Z-scores were calculated based on the mean and standard deviation of a reference set (control without serum, unless otherwise stated). Hierarchical clustering based on Pearson's correlation was performed on the log-transformed normalized data after median centering per metabolite. The multivariate statistical analysis included independent component analysis (ICA) (http://metagenealyse.mpimp-golm.mpg.de/). ICA was used to discriminate sample patterns, to identify the metabolites associated with infection and to minimize the interindividual variation's influence. Principal component analysis (PCA) was used to reduce the high dimension of the data set.

**Measurement of NADH and cAMP concentration**. NADH was measured with a commercial kit according to the manufacturer's instruction (Cat. GMS60097.2, Genmed, Shanghai, China). Bacteria with 3 × 10⁹ CFU were disrupted by sonication (20% output and 5 s interval time for a total of 3 min) in an ice-bath. For measurement of NADH, the samples were centrifuged at 12,000 × *g* for 10 min at 4 °C. Supernatants were collected and protein concentration was measured. A total of 0.3 μg of protein were used for NADH quantification. For measurement of cAMP, the samples were centrifuged at 12,000 × *g* for 5 min at 4 °C, and 100 μL supernatant of each sample was used for measurement according to the manufacturer's instructions (Cat. K371, cAMP Direct Immunoassay kit (Colorimetric), BioVision, USA).

**Membrane potential**. *E. coli* cells were adjusted to 1 × 10⁶ cells/mL in saline, and with 10 μM DiOC2(3) for 30 min at 37 °C. One milliliter of cultures was analyzed on a FACSCalibur flow cytometer at 37 °C (Becton Dickinson, San Jose, CA, USA). Each sample was observed with forwarding versus side scatter and was gated before data acquisition, record count: 50,000 cells. The fluorescence of DIOC₂(3)'s can be excited at 488 nm and emitted at 530 nm (green) or 610 nm (red). The green fluorescence is determined by size while the red fluorescence is determined by membrane potential. The ration of red to green indicated fluorescence intensity values of the gated populations. The computational formula of membrane potential = Log(10^{3/2} × red fluorescence/green fluorescence)[52].

**Absolute ATP and α-Ketoglutarate levels and ATPase activity**. Intracellular water content was measured as follows[53]. Bacterial cells were harvested in the OD₆₀₀ of 1.0 by centrifugation for 10 min at 12,000 × *g* and washed twice by centrifugation with sterile saline. Followed by measurement of the wet and dry weight of the cells, the intracellular water content/g was calculated from the equation: 1−0.23 - *D/F*, where *F* is the wet weight and *D* is dry weight. The constant value is 0.23.

To quantify intracellular ATP, α-Ketoglutarate concentrations and F type ATPase activity, dry cells from 10 mL of 1.0 OD₆₀₀ were dissolved in 100 μL PBS buffer (pH7.4) and disrupted by sonication (20% output and 5 s interval time for a total of 3 min) in an ice-bath. The solution was diluted 1:500 for measurement of ATP and α-Ketoglutarate concentrations and F-type ATPase activity, with BacTiter-Glo™ Microbial Cell Viability Assay (Cat. G8231, Promega, Madison, WI), α-Ketoglutarate Colorimetric/Fluorometric Assay Kit (Cat. K677, BioVision, USA) and F-type ATPase Activity Assay Kit (Cat. GMS50248.2, Genmed, Shanghai, China), respectively. The intracellular concentrations and enzyme activity unit were quantified according to the manufacturer's manual and intracellular water content[39,53].

**Western blot, far-Western blot, and Co-IP**. Western blot, far-Western blot, and Co-IP were performed as follows[54–56]. For Western blot, samples were dissolved in 15% SDS-PAGE. Rabbit polyclonal antibodies anti-HtrE, -NfrA, -YhcD, -GlyA, -Kbl, -PurD, -CRP (1:500 dilution) were used as primary antibodies, which came from our laboratory and validated by corresponding mutants. Horseradish peroxidase (HRP)-mouse anti-ribbit IgG (1:3000 dilution, Boson Biotech, Xiamen China) was used as the secondary antibody. For far-Western blot, recombinant proteins NfrA, HtrE YhcD on nitrocellulose membrane (NC) membranes were used as bait proteins to capture human serum complement C3b or IgG (1:100 dilution) and then anti-C3b or anti-IgG were used to recognize them, respectively. For Co-IP, beads coupled with C3b antibody (Hycult Biotech Inc., PA, USA) were incubated with a human serum to capture C3b as a bait protein. The bait protein was reacted with recombinant proteins NfrA, HtrE, and YhcD. Recovered pellets were isolated by SDS/PAGE and transferred to NC membranes and then recognized by corresponding antibodies anti- HtrE, -NfrA, or -YhcD. The membranes were developed using electro-chemi-luminescence ECL. Uncropped and unprocessed scans of all blots are attached in the Source Data file or as supplementary figures in the Supplementary Information.

**Detection of O-antigens**. Diagnostic sera for Entero-toxigenic *Escherichia coli* (Cat. TR301, TR302 & TR303) were commercially obtained from Ningbo Tianrun Bio-pharmaceutical Co. LTD, Ningbo, China. O-antigens were detected according to the manufacturer's instruction. Briefly, two lines were drawn on a clean glass slide with a crayon to form two cells. In sterile conditions, one drop of diagnosed serum and sterile saline were added in the first cell as a control and the second cell as a test, respectively. 1 × 10⁷ CFU *E. coli* were mixed in the second cell and then mixed in the first cell until these bacteria were mixed with saline or serum to evenly form an emulsion. The glass slides were gently shaken for 1–2 min and observed with the naked eye. A positive reaction was determined when a milky white agglutination appeared, while a negative reaction resulted from a still equal emulsion. The results could be confirmed under a low power microscope.

**Generation of gene-deletion mutants**. Knockout of Y3-*glyA*, Y21-*glyA*, Y1-*waal*, Y3-*waal*, Y7-*waal*, Y10-*waal*, Y12-*waal*, Y17-*waal*, Y20-*waal*, Y23-*waal*, Y25-*waal*, Y26-*waal*, Y28-*waal*, Y29-*waal*, Δ*nfrA*Δ*yhcD*, Δ*htrE*Δ*yhcD*, Δ*yhcD*Δ*htrE*, Δ*nfrA*-Δ*yhcD*Δ*htrE*, Δ*htrE*Δ*yhcD*Δ*nfrA*, and Δ*yhcD*Δ*htrE*Δ*nfrA* was performed with one-step inactivation of chromosomal genes. For the construction of these mutants,

primers used to clone the kanamycin cassette from pKD13 are listed in Supplementary Table 6. The PCR products were transformed into targeted strains bearing pSIM5 plasmid, which express lambda red recombinase (*E. coli* Y3 and Y21 for *glyA*-deletion, Y1, Y3, Y7, Y10, Y12, Y17, Y20, Y23, Y25, Y26, Y28, Y29 for *waal*-deletion, Δ*nfrA*, Δ*htrE* and Δ*yhcD* for *yhcD*-, *yhcD*- and *htrE*-deletion, and Δ*nfrA*Δ*yhcD*, Δ*htrE*Δ*yhcD*, Δ*yhcD*Δ*htrE* and for *htrE*-, *nfrA*- and *nfrA*-deletion, respectively). The transformants were selected on 50 µg/mL kanamycin. The deletion mutants were confirmed by PCR. Test primers are listed in Supplementary Table 5. Kanamycin cassette was removed by transforming pCP20 plasmid.

**Detection of C3b/iC3b, C3, C5, and C9 neoantigen.** Fluorescein isothiocyanate (FITC) conjugated monoclonal antibody to human complement factor C3b/iC3b (Cat. HM2286, clone 3E7) or a C9 neoantigen of the terminal complement complex (TCC) (Cat.HM2167F, clone aE11) were from Hycult Biotech Inc., PA, USA. Anti-C3 Rabbit pAb (Cat. 381678), anti-C5 (N-term) Rabbit pAb (Cat. 616722), and Fluorescein (FITC) affiniPure goat anti-rabbit IgG (Cat.511101) were from Zenbio, China. For detection of C3b/iC3b and C9 neoantigen, $10^6$ CFU/mL diluted cells were mixed with 1 µg C3b/iC3b FITC-monoclonal antibody or 0.5 µg FITC-C9 neoantigen monoclonal antibody for 1 h at 37 °C. For detection of C3 and C5, $10^6$ CFU/mL diluted cells were mixed with 1:20 C3 polyclonal antibody or 1:20 C5 (N-term) polyclonal antibody for 1 h at 37 °C. Unbound primary antibody was removed and then mixed with Fluorescein (FITC) affiniPure goat anti-rabbit IgG at 1:300 for 1 h at 37 °C. Aliquots 1 mL of culture was added into flow tubes and then analyzed on a FACSCalibur flow cytometer. Each sample was observed with forwarding versus side scatter and was gated before data acquisition, record count: 50,000 cells. Experiments were repeated in at least three independent biological replicates.

**Isotope tracer experiment.** Stable isotope-labeled compounds were detected by non-targeted detection[37,38]. *E. coli* K12 was cultured using an unlabeled glycine and a 1:1 mixture unlabeled glycine and [U-13C₂] glycine. Three independent biological replicates were performed for each condition. GC-MS was performed using an Agilent 7890A GC equipped with an Agilent 5975C MS operating under electron impact (EI) ionization. The effectiveness of each tracer was gauged using the software downloaded from Internet[57].

**Protein-small molecular interaction.** MST was carried out as follows[58]. AtpB, PurR, NfrA, HtrE or YhcD and the N-Hydroxysuccinimide (NHS) ester fluorescent dye solutions were mixed in 1:1 ratio and incubated for 30 min at room temperature in the dark. Serial dilutions of unlabeled α-Ketoglutarate, *glyA_p* and *kbl_p* and human complement C3 with PBS buffer were mixed with 20 µM of fluorescent-labeled AtpB (for AtpB-α-Ketoglutarate), PurR (for PurR-*glyA_p* and -*kbl_p*) and NfrA, HtrE or YhcD (for human complement C3) and then incubated for 5 min at room temperature. Sample 4 µL was loaded into MST-glass capillaries for MST-analysis in a NanoTemper Monolith NT.115. By plotting α-Ketoglutarate, *glyA_p*, *kbl_p* or human complement C3 concentration to ‰ changes of normalized fluorescence (△Fnorm [‰]), curve fitting was performed using GraphPad Prism software and *K*d values were determined.

**Electrophoretic mobility shift assay (EMSA).** Purified recombinant PurR and *glyA* promoter (212 bp) or *kbl* promoter (208 bp) were incubated in the binding buffer consisting of 25 mM Tris (pH 6.8), 150 mM NaCl, 5 mM EDTA, 10 mM DTT, 50% vol/vol glycerol. Electrophoresis was performed in an 8% polyacrylamide (29:1 acrylamide:bisacrylamide) gel, casted in 20 mM Tris–borate, 1 mM EDTA (pH 8.0), 1% (wt/vol) ammonium persulphate, 0.5% (vol/vol) TEMED and run in 20 mM Tris–borate, 1 mM EDTA (pH8.0) at 4 °C at 80 V and stained with a solution of 5 µg/mL ethidium bromide (EB).

**Bacterial clearness in a mouse model.** B6/JNju-Rag1em2Cd14/Nju mouse (six weeks), genotype:(Rag1)KO/KO were obtained from Nanjing Biomedical Research Institute of Nanjing University, Nanjing, China. BALB/c mice (8–10 weeks) were obtained from the Experimental Animal Center of Sun Yat-sen University, Guangzhou, China. These animals were acclimatized for one week and randomly divided into control and experimental groups (*n* = 6). The experimental groups were intraperitoneally injected with 10, 15 and 20 mg glycine/mouse, and control was injected with equal volume sterile saline, twice a day. After 2 days, $10^6$ CFU *E. coli* K12, Δ*cycA*, Δ*htrE-nfrA-yhcD*, *P. aeruginosa*, MRSA or *K. pneumonia*/mouse were injected. Blood was withdrawn at 24 h and 48 h (for *E. coli* dynamic observation), and spleen, kidney, and liver were collected at 24 h after the last injection to spot-plate and to observe CFU. Only those of dilutions yielding 20–200 colonies were enumerated to calculate CFU. During the experiment, no side effects were found in these animals. Animal protocols were approved by the University committee for the use and care of laboratory animals.

**Glycine-potentiated killing by ampicillin in a mouse model.** Six weeks BALB/c mice were obtained from the experimental animal center of Sun Yat-sen University, Guangzhou, China and were acclimatized for one week. These animals were intraperitoneally injected with of clinic *E. coli* Y9 (1.5 × $10^8$ CFU), *E. coli* Y23

(2 × $10^8$ CFU), *P. aeruginosa* P22 (5 × $10^5$ CFU) or *K. peneumoniae* 0367 (2 × $10^7$ CFU) per mouse by intravenous injection. The mice were randomly divided into four groups, ten per group and were intraperitoneally injected with saline (group 1), 800 mg kg$^{-1}$ glycine (group 2), 320 mg kg$^{-1}$ ampicillin (group 3) and glycine (800 mg kg$^{-1}$) plus 320 mg kg$^{-1}$ ampicillin (group 4). The injections were performed at 1 h, 36 h and 48 h post the infection, mouse survival was monitored for fifteen days.

**UPLC-MS/MS to detect glycine in blood and internal organs.** To determine glycine concentration, plasma was drawn from mice and mixed immediately with acetonitrile (1:3, v/v). The mixture was centrifuged twice at 12,000 × *g* for 10 min, and supernatants were collected. To determine glycine concentrations in liver, kidney, and spleen, these internal organs were removed from 10 untreated mice and 10 treated mice and weighted. These organs were homogenized in acetonitrile (1:2, w/v). Followed by centrifugation twice at 12,000 × *g* for 10 min, supernatants were collected. The supernatants from both plasma and organs were filtered through a 0.22 µM pore size hydrophilic PVDF membrane. Liquid chromatography was performed at 35 °C using a Waters ACQUITY UPLC system equipped with an Acquity BEH C$_{18}$ column (50 mm × 2.1 mm i.d., 1.7 µm; Waters Corp.). Separation was using linear gradient elution with mobile phase A (acetonitrile) and B (0.2% formic acid in ultra-pure water) at a flow rate of 0.2 mL/min. The gradient elution was as follows: 0–3 min, 40% A and 60% B. The injection volume was 3 µL. Mass spectrometry experiments and optimization were carried out with QUATTRO PREMIER XE equipped with an electrospray ionization source operating in positive ionization mode (ESI+). The capillary voltage was set to 3000 V; the cone voltage was set to 20 V. The extractor voltage and RF Lens were set at 3 V and 0.5 V, respectively. The desolvation gas flow was set to 650 L/h at a temperature of 450 °C, the cone gas flow rate was set at 50 L/h and the source temperature was set at 120 °C.

**Gene complementation.** The primers used for amplification are listed Supplementary Table 7. The deleted genes were amplified by PCR and cloned into the *Hind* III and *BamH* I sites of the pACYC184 Cm(+)vector or were constructed homologous recombination using pACYC184 Cm(+)vector. The recombinant plasmids were introduced into DH5α competent cells by electroporation under the condition of 2.5 kV voltage and 6 ms pulse time. The samples were incubated at 37 °C for 45 min and spread onto LB plates containing 50 µg/ml chloramphenicol at 37 °C. Recombinant plasmids were checked by digestion with restriction endonucleases and electroporated into *E. coli* deletion cells. The acquisition of the deletion mutants utilized a kanamycin-resistant gene to replace these genes by homologous recombination. Thus, these genes and their upstream and downstream of about 200 bp sequences were used for designation of primers. The resulting PCR products were approximately 1700 bp, containing kanamycin-resistant gene sequence (1303 bp), upstream (200 bp) and downstream (200 bp).

**Reverse-transcription qPCR.** Total RNA samples were prepared from indicated tissues using TRIzol reagent (Invitrogen). Double-stranded cDNA was synthesized from total RNA using the SYBR Perfect real-time series kits (Takara). cDNA was analyzed by real-time RT-PCR using Roche's LightCycler 480 real-time PCR system. Quantitative PCR was performed on each cDNA sample in triplicate. 16s rRNA was used as an internal control to normalize expression levels. Expression level was calculated using the comparative 2-ΔΔCt method. All samples were analyzed in three independent biological replicates with results expressed as the mean ± SEM[59]. The primers used for amplification are listed in Supplementary Table 8.

**PCR and sequencing of waal.** PCR was carried out in a routine procedure. Primers were listed in Supplementary Table 9. 50 µL reaction system contains 2 µl template DNA, 1 µL 10 µM upper or down primer, 2.0 µL 2.5 mM dNTP mix (TAKARA), 5 µL 10 × PCR polymerase buffer (TAKARA), 0.5 µL Taq polymerase (TAKARA) and appropriate amount of ddH$_2$O. The PCR programs were applied as following: the denaturation temperature 94 °C for 3 min and 94 °C for 40 s; annealing temperature 56 °C for 30 s; extension at 72 °C for 90 s; and final extension 72 °C for 5 min; with 35 cycles of denaturation, annealing and extension phases. PCR products were checked with 0.8% agarose gel and were sequenced (BGI, Shenzhen, China).

**Reporting summary.** Further information on research design is available in the Nature Research Reporting Summary linked to this article.

## Data availability
All relevant data are available from the authors upon reasonable requests. The source data underlying all Main and Supplementary Figures and Tables are provided as a Source Data file.

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

## Acknowledgements

This work was sponsored by grants from the National Key Research and Development Program of China (2018YFD0900501/4), NSFC projects (U1701235, 31822058, 31770045, 41276145), and The Blue Life Breakthrough Program of LMBB for Pilot National Laboratory, Marine Science and Technology (Qingdao), China (MS2018NO06).

## Author contributions

X.X.P., B.P. and H.L. wrote the manuscript. X.X.P., B.P., H.L. and T.T.Z. conceptualized and designed the project. Z.G.C., X.X.P., T.T.Z., B.P., H.L., Z.X.C., C.G., T.C.Y. and J.Y.Z. interpreted the data. C.G., Z.G.C., J.Y.Z., T.T.Z., T.C.Y. and J.X.Z. performed data analysis and collected samples. Z.X.C., H.L., J.W. and D.L. performed experiments.

## Additional information

**Competing interests:** The authors declare no competing interests.

