## [Peer Review File · Nature Communications]

Reviewers' comments:

Reviewer #1 (Remarks to the Author):

In the manuscript titled "Glycine, serine, and threonine metabolism confounds efficacy of complement-mediated killing", Cheng et al explore the metabolic mechanisms underlying bacterial serum sensitivity or resistance. More specifically, the authors determine that exposure of *E. coli* to serum (and by extension, complement) leads to global metabolic changes, including significant down regulation of glycine, serine, and threonine metabolism. In a series of comprehensive and well-controlled experiments, the authors go on to demonstrate that mechanistically, this alteration of amino acid catabolism is linked to changes in TCA cycle flux, ATP production, membrane potential, cAMP/CRP regulation, and finally, cell surface protein expression. It is the alterations in cell surface protein expression that at least partially underlie the altered susceptibility to serum killing, as the authors clearly demonstrate decreased binding of complement factors. The authors also show that exogenous glycine, serine, and threonine can restore serum-sensitivity to serum-resistant *E. coli*, and that glycine treatment sensitizes many classes of human bacterial pathogens to serum. Finally, the authors show that mice treated with glycine have significantly lower bacterial burdens during disseminated infection with 4 different pathogens. Taken together, these experiments outline a new mechanistic understanding of bacterial serum sensitivity and suggest a potential avenue for modulating immune killing of common bacterial pathogens. The work is broad in scope, comprehensively executed, and will be of significant interest to the field of investigators studying innate immunology, bacterial pathogenesis, and host-pathogen interactions. The conclusions are soundly supported by the data, and the authors very comprehensively address potential confounding variables. Statistical analyses are rigorously applied throughout the manuscript. I do not believe that additional experiments are necessary to fortify the main conclusions. However, some items should be addressed in the text prior to publication.

Minor comments:

1. There are grammatical errors throughout. The manuscript may benefit from a professional editor in this regard
2. Lines 184-186: This statement is likely worded too strongly. Treatment with ampicillin and gentamicin did not potentiate glycine killing, but this certainly doesn't completely exclude changes in membrane integrity. To make this statement conclusively would require additional compounds that more directly test membrane integrity, or perhaps EM.
3. Lines 284 and 285: The authors note that alanine, fructose, glucose, and glutamine stimulate the TCA cycle and increase membrane potential, but do not decrease survival in the presence of serum. This (and other comments below) point to a multi-factorial mechanism that may include more than just changes in membrane potential or altered surface protein expression. The authors should elaborate on this in the discussion. For example, why is MRSA killing potentiated by glycine in the

absence of the cell surface proteins that bind complement in *E. coli*? What are some alternative hypotheses as to why glycine is potentiating killing, but other metabolites that alter membrane potential do not?

4. Lines 315-316: I believe this statement is worded wrong. Doesn't a decrease in glycine lead to an increase in killing?

5. Figure 3, panel f: Please define the green dotted line

6. Line 336: Suggest replacing "genes" with "the presence of genes" or "gene expression"

7. Figure 4, panel D: Ratio is misspelled as ration

8. Figure 5: Please define ATPgammaS in the legend

9. Figure 6, panel d: italicize *flhC*

10. Figure 6, panel j: This graph shows that the surface proteins are clearly only part of the reason for increased serum sensitivity. Please elaborate on this in the discussion.

11. Line 576: change "explore" to "explored"

12. Lines 638-642 seem out of place in this data section. consider moving

13. Figure 8, panel B: MRSA is spelled wrong

14. Discussion: Please discuss how you think glycine is potentiating serum killing of the non-*E. coli* strains. I understand that this may be speculative.

15. Line 1076: Stating that bacterial strains "were from collections in our laboratory" does not provide enough information to the reader to enable reproducibility. Did these strains come from published work? Were they obtained from patients? Please add a strain table and appropriate references.

Jim Cassat

Reviewer #2 (Remarks to the Author):

In this paper by Cheng et. al, the influence and mechanisms of glycine, serine and threonine on serum-mediated killing of bacteria is investigated. The paper is thorough and contains novel research of great interest that could have real world consequences for treatment of bacteremia. The authors were rigorous in their experimental design, with mostly good controls used throughout. The

extension into clinically relevant bacterial strains and mouse model was also excellent. The paper is overall convincing however have a few major concerns as outlined below:

Major comments:

- As stated in the introduction, LPS is one of the main drivers of serum resistance- primarily due to the production of O-antigen that is thought to block access of complement to the surface/ outer membrane proteins of the bacteria. K-12 has a mutation in waaL which renders it O-antigen negative and thus its role in any glycine mediated sensitivity is unexplored. It is likely that many of the other strains used in this study are O-antigen positive however this is not mentioned. Thus,

1) As the second most used strain- Is the serum-resistant Y17 O-antigen positive?

2) Is it known if any of the other E. coli, Pseudomonas etc. used were O-antigen positive?

3) Was the effect of glycine on O-antigen expression investigated at all?

4) As we know from the serum resistome paper that O-antigen expression in EC958 is critical could you test the effect of glycine on killing of this strain?

- Gram negative bacteria are sensitive to complement-mediated killing as the MAC complex can insert into the outer membrane. Gram positive bacteria however are resistant to complement mediated killing as the peptidoglycan of their membrane is too thick for MAC insertion. Thus,

5) In figure 8A All Staph aureus strain (Gram positive) are killing by the addition of glycine to serum. By what mechanism is this killing happening? Was a HI-serum (or C3 depleted) + glycine control used in these assays? In fact – was this control done for any of the strains in 8A?

Minor comments:

1) In the introduction there is no mention of the role of capsule in serum resistance (although reports differ). Also no mention of antibody mediated serum resistance found in Gram negatives such as Salmonella (MacLennan et al. Science 2010), Pseudomonas (Wells et al 2014 JEM) and E. coli (Coggon et al. 2018 MBio).

2) Fig 2b- no mention of what strain is used and text suggests could be Y17

3) In many places I am not sure if stats are just missing or there was no significant difference even though graph and error bars look different; Fig 3d, Fig 8b, Sup Fig 5A, Sup Fig 14

4) Supplementary Figure 5A would be easier to interpret if the glycine minus results (in figure 4) were also put into the same graph. Could these figures be combined?

Responses to Reviewer's comments one by one

Reviewers' comments:

Reviewer #1 (Remarks to the Author):

In the manuscript titled "Glycine, serine, and threonine metabolism confounds efficacy of complement-mediated killing", Cheng et al explore the metabolic mechanisms underlying bacterial serum sensitivity or resistance. More specifically, the authors determine that exposure of *E. coli* to serum (and by extension, complement) leads to global metabolic changes, including significant down regulation of glycine, serine, and threonine metabolism. In a series of comprehensive and well-controlled experiments, the authors go on to demonstrate that mechanistically, this alteration of amino acid catabolism is linked to changes in TCA cycle flux, ATP production, membrane potential, cAMP/CRP regulation, and finally, cell surface protein expression. It is the alterations in cell surface protein expression that at least partially underlie the altered susceptibility to serum killing, as the authors clearly demonstrate decreased binding of complement factors. The authors also show that exogenous glycine, serine, and threonine can restore serum-sensitivity to serum-resistant *E. coli*, and that glycine treatment sensitizes many classes of human bacterial pathogens to serum. Finally, the authors show that mice treated with glycine have significantly lower bacterial burdens during disseminated infection with 4

different pathogens. Taken together, these experiments outline a new mechanistic understanding of bacterial serum sensitivity and suggest a potential avenue for modulating immune killing of common bacterial pathogens. The work is broad in scope, comprehensively executed, and will be of significant interest to the field of investigators studying innate immunology, bacterial pathogenesis, and host-pathogen interactions. The conclusions are soundly supported by the data, and the authors very comprehensively address potential confounding variables. Statistical analyses are rigorously applied throughout the manuscript. I do not believe that additional experiments are necessary to fortify the main conclusions. However, some items should be addressed in the text prior to publication.

Response. Thank you for your comment!

Minor comments:

1. There are grammatical errors throughout. The manuscript may benefit from a professional editor in this regard

Response, We appreciate this comment! The manuscript has been carefully checked and revised by professional editor.

2. Lines 184-186: This statement is likely worded too strongly. Treatment with ampicillin and gentamicin did not potentiate glycine killing, but this certainly doesn't completely exclude changes in membrane integrity. To make this statement conclusively would require additional compounds that more directly test membrane

integrity, or perhaps EM.

Response, We appreciate this comment! According to Reviewer’s comment, it has been revised in lines 192 – 193 following as “This result indicates that glycine-mediated serum killing may not be related to impaired bacterial membrane integrity.”.

3. Lines 284 and 285: The authors note that alanine, fructose, glucose, and glutamine stimulate the TCA cycle and increase membrane potential, but do not decrease survival in the presence of serum. This (and other comments below) point to a multi-factorial mechanism that may include more than just changes in membrane potential or altered surface protein expression. The authors should elaborate on this in the discussion. For example, why is MRSA killing potentiated by glycine in the absence of the cell surface proteins that bind complement in *E. coli*? What are some alternative hypotheses as to why glycine is potentiating killing, but other metabolites that alter membrane potential do not?

Response, According to Reviewer’s comment, it has been revised in lines 727 –740 as following “Notably, other metabolites like alanine, fructose, glucose, and glutamine promote the membrane potential, but they fail to reverse serum resistance. This discrepancy is attributed to the dual roles of glycine that not only promotes membrane potential but also up-regulates the expression of complement-binding outer membrane proteins. The proposed mechanism can be applicable in Gram-negative bacteria with outer membrane proteins in addition to *E. coli*, but why this effect is also

observed in Gram-positive bacteria requires further investigation as they do not possess the outer membrane proteins for MAC insertion. The increased deposition of complement components on MRSA surface suggests other mechanisms may exist. Nevertheless, that the triple mutations of *htrE*, *nfrA* and *yhcD* did not completely abrogate glycine-mediated serum killing and loss of *waal* affected the killing potential in several mutants indicates that serum resistance is a multi-factorial event that may include more than membrane potential and outer membrane proteins.”

4. Lines 315-316: I believe this statement is worded wrong. Doesn't a decrease in glycine lead to an increase in killing?

Response, We appreciate this comment! It has been revised in lines 313 – 314 as following “The glycine concentrations were proportional to the percent survival of the mutants”.

5. Figure 3, panel f: Please define the green dotted line

Response, We appreciate this comment! According to reviewer’s comment, it has been defined in line 326 – 327 as following “The green dotted line indicated the average level of glycine in *E. coli* K12.

6. Line 336: Suggest replacing "genes" with "the presence of genes" or "gene expression"

Response, We appreciate this comment! According Reviewer’s comment, it has been

revised as “the presence of genes”.

7. Figure 4, panel D: Ratio is misspelled as ration

Response, We appreciate this comment! It has been corrected.

8. Figure 5: Please define ATPgammaS in the legend

Response, We appreciate this comment! According Reviewer’s comment, it has been defined in lines 440 as following “ATP γ S, a nonhydrolyzable ATP analog.”.

9. Figure 6, panel d: italicize flhC

Response, We appreciate this comment! It has been revised.

10. Figure 6, panel j: This graph shows that the surface proteins are clearly only part of the reason for increased serum sensitivity. Please elaborate on this in the discussion.

Response, We appreciate this comment! According Reviewer’s comment, we have discussed this in lines 736 - 740 as following “Nevertheless, that the triple mutations of *htrE*, *nfrA* and *yhcD* did not completely abrogate glycine-mediated serum killing and loss of *waal* affected the killing potential in several mutants indicates that serum resistance is a multi-factorial event that may include more than membrane potential and outer membrane proteins.”.

11. Line 576: change "explore" to "explored"

Response, We appreciate this comment! It has been revised.

12. Lines 638-642 seem out of place in this data section. consider moving

Response, We appreciate this comment! They have been removed.

13. Figure 8, panel B: MRSA is spelled wrong

Response, We appreciate this comment! It has been corrected.

14. Discussion: Please discuss how you think glycine is potentiating serum killing of the non-E. coli strains. I understand that this may be speculative.

Response, We appreciate this comment! We have discussed this issue in the revised manuscript in lines 729 - 733 as following “This discrepancy is attributed to the dual roles of glycine that not only promotes membrane potential but also up-regulates the expression of complement-binding outer membrane proteins. The proposed mechanism can be applicable in Gram-negative bacteria with outer membrane proteins in addition to *E. coli*”.

15. Line 1076: Stating that bacterial strains "were from collections in our laboratory" does not provide enough information to the reader to enable reproducibility. Did these strains come from published work? Were they obtained from patients? Please add a strain table and appropriate references.

Response, We appreciate this comment! The strains were listed in **Supplementary**

Tab. 5.

Supplementary Tab. 5 Bacterial strains used in the study

Strain	Original	Strain	Original	Strain	Original
E.coli K12¹	KEIO collection	Δ sdhD	KEIO collection	Δ sdaA	KEIO collection
Δ nfrA	KEIO collection	Δ frdA	KEIO collection	Δ sdaB	KEIO collection
Δ htrE	KEIO collection	Δ frdB	KEIO collection	Δ tdcG	KEIO collection
Δ yhcD	KEIO collection	Δ frdB	KEIO collection	Δ trpA	KEIO collection
Δ purR	KEIO collection	Δ frdC	KEIO collection	Δ trpB	KEIO collection
Δ fis	KEIO collection	Δ frdD	KEIO collection	Δ tynA	KEIO collection
Δ crp	KEIO collection	Δ mdh	KEIO collection	Δ aecE	KEIO collection
Δ cyaA	KEIO collection	Δ mgo	KEIO collection	Δ aecF	KEIO collection
Δ flhC	KEIO collection	Δ purL	KEIO collection	Δ gltA	KEIO collection
Δ csgD	KEIO collection	Δ purK	KEIO collection	Δ acnA	KEIO collection
Δ atpA	KEIO collection	Δ purE	KEIO collection	Δ acnB	KEIO collection
Δ atpC	KEIO collection	Δ purC	KEIO collection	Δ ybhJ	KEIO collection
Δ atpD	KEIO collection	Δ purH	KEIO collection	Δ icd	KEIO collection
Δ glyA	KEIO collection	Δ guaA	KEIO collection	Δ sucA	KEIO collection
Δ kbl	KEIO collection	Δ guaB	KEIO collection	Δ sucB	KEIO collection
Δ purD	KEIO collection	Δ guaC	KEIO collection	Δ sucC	KEIO collection

$\Delta ltaE$	KEIO collection	Δndk	KEIO collection	$\Delta sucD$	KEIO collection
$\Delta ilvA$	KEIO collection	$\Delta mazG$	KEIO collection	$\Delta sdhA$	KEIO collection
$\Delta tdcB$	KEIO collection	Δgsk	KEIO collection	$\Delta sdhB$	KEIO collection
$\Delta pepA$	KEIO collection	$\Delta ushA$	KEIO collection	$\Delta sdhC$	KEIO collection
$\Delta pepB$	KEIO collection	$\Delta yibR$	KEIO collection	$\Delta pykA$	KEIO collection
$\Delta pepN$	KEIO collection	$\Delta surE$	KEIO collection	$\Delta pykF$	KEIO collection
$\Delta gcvP$	KEIO collection	Δyjg	KEIO collection	$\Delta gshB$	KEIO collection
Δtdh	KEIO collection	Y3 $\Delta glyA$	This study	Y17 $\Delta waal$	This study
$\Delta nfrA\Delta yhcD$	This study	Y21 $\Delta glyA$	This study	Y20 $\Delta waal$	This study
$\Delta htrE\Delta yhcD$	This study	Y1 $\Delta waal$	This study	Y23 $\Delta waal$	This study
$\Delta yhcD\Delta htrE$	This study	Y3 $\Delta waal$	This study	Y25 $\Delta waal$	This study
$\Delta nfrA\Delta yhcD\Delta htrE$	This study	Y7 $\Delta waal$	This study	Y26 $\Delta waal$	This study
$\Delta htrE\Delta yhcD\Delta nfrA$	This study	Y10 $\Delta waal$	This study	Y28 $\Delta waal$	This study
$\Delta yhcD\Delta htrE\Delta nfrA$	This study	Y12 $\Delta waal$	This study	Y29 $\Delta waal$	This study
Vibro. alginolyticus			The collections of our laboratory ²		
Vibro. parahaemolyticus			The collections of our laboratory		
E.coli O157:H7			The collections of our laboratory ³		
30 strains of multidrug-resistant					
Pseudomonas aeruginosa			The collections of our laboratory, which were isolated from patients		
30 strains of methicillin-resistant					
Staphylococcus aureus (MRSA)			The collections of our laboratory, which were isolated from patients		
30 strains of multidrug-resistant Klebsiella			The collections of our laboratory, which were isolated from patients		

pneumonia

30 strains of multidrug-resistant *E.coli*

The collections of our laboratory, which were isolated from patients

¹Baba T, et al. Construction of *Escherichia coli* K-12 in-frame, single-gene knockout mutants: the Keio collection. *Mol Syst Biol.* 2, 2006.0008 (2006).

²Xiong XP, Wang C, Ye MZ, Yang TC, Peng XX, Li H. Differentially expressed outer membrane proteins of *Vibrio alginolyticus* in response to six types of antibiotics. *Mar Biotechnol*, 2010, 12:686-695

³Li H, Xiong XP, Peng B, Xu CX, Ye MZ, Yang TC, Wang SY, Peng XX. Identification of broad cross-protective immunogens using heterogeneous antiserum-based immunoproteomic approach. *J Proteome Res.*, 2009, 8, 4342-4349

Reviewer #2 (Remarks to the Author):

In this paper by Cheng et. al, the influence and mechanisms of glycine, serine and threonine on serum-mediated killing of bacteria is investigated. The paper is thorough and contains novel research of great interest that could have real world consequences for treatment of bacteremia. The authors were rigorous in their experimental design, with mostly good controls used throughout. The extension into clinically relevant bacterial strains and mouse model was also excellent. The paper is overall convincing however have a few major concerns as outlined below:

Response, We appreciate this comment!

Major comments:

- As stated in the introduction, LPS is one of the main drivers of serum resistance- primarily due to the production of O-antigen that is thought to block access of complement to the surface/ outer membrane proteins of the bacteria. K-12 has a mutation in waaL which renders it O-antigen negative and thus its role in any glycine mediated sensitivity is unexplored. It is likely that many of the other strains used in this study are O-antigen positive however this is not mentioned. Thus,

1) As the second most used strain- Is the serum-resistant Y17 O-antigen positive?

2) Is it known if any of the other E. coli, Pseudomonas etc. used were O-antigen positive?

3) Was the effect of glycine on O-antigen expression investigated at all?

4) As we know from the serum resistome paper that O-antigen expression in EC958 is critical could you test the effect of glycine on killing of this strain?

Response: We appreciate this comment! According to Reviewer's comments, these experiments have been performed to further clarify the glycine-mediated serum killing. As you suggest to use the EC598, we have tried very hard to request this strain but fail, possibly due to the limited time. In this regard, we have added the result of O157:H7.

Experiments to test whether the other strains in the study are O-antigen positive and

the effect of glycine on O-antigen expression and strain killing as **Figs. 6p - r**, **Supplementary Tab. 3** and **Fig. 14** and described in lines 533 – 546 as following “**In addition, we investigated effect of O-antigen on the glycine-mediated killing, which is critical for LPS-mediated serum resistance in *E. coli*^{10, 19-21}. Among the 30 clinical *E. coli* strains, 27 strains were O-antigen positive (**Supplementary Tab. 3**). Therefore, *Waal*, catalyzing the transfer of O-antigen polysaccharide from its lipid-linked intermediate to a terminal sugar of the lipid A-core oligosaccharide during LPS biosynthesis, is subjected to frequent mutations to generate serum resistance⁴⁷. The mutations of *waal* were detected in randomly selected *E. coli*, *P. aeruginosa*, and *K. pneumonia* strains (**Supplementary Fig. 14**). *E. coli* serotype O157:H7, with adhesive fimbriae and a cell wall that consists of an outer membrane containing LPS, was sensitive to the glycine-triggering killing (**Fig. 6p**) while exogenous glycine did not affect *waal* transcription (**Fig. 6q**). When *waal* was deleted, the glycine-mediated killing was either unaffected or reduced by 4.81-24.1% (**Fig. 6r**). These results indicate that LPS is not the key target in the glycine-triggering serum killing.”**

Fig. 6 p, Percent survival of *E. coli* O157:H7 in the presence or absence of 100 μ L serum or 100 mM glycine or both. **q**, qRT-PCR for detection of *waal* expression in

the presence or absence of 100 mM glycine. **r**, Percent survival of the indicated *E. coli* strains and their genetical mutants with *waal* deletion in the presence of 100 μ L serum and 100 mM glycine.

Supplementary Tab. 3 O-antigen distribution of clinical isolated strains

Strains	O-antigen	Strains	O-antigen	Strains	O-antigen
Y1	O55:K59	Y11	O29:K?	Y21	O142:K86
Y2	O124:K72	Y12	O152:K?	Y22	O125:K70
Y3	O119:K69	Y13	O6:K15	Y23	
Y4	O127a:K63	Y14		Y24	O144:K?
Y5		Y15	O119:K69	Y25	O112:K66
Y6	O28:K73	Y16	O25:K19	Y26	O20:K17
Y7	O8:K40	Y17	O126:K71	Y27	O152:K?
Y8	O143:K?	Y18	O9:K9	Y28	O128:K67
Y9	O78:K80	Y19	O29:K?	Y29	O26:K60
Y10	O119:K69	Y20	O136:K78	Y30	O7:K1

Note: ? is marked by the Kit, indicating it is not exactly identified.

Supplementary Fig. 14 PCR (a) and sequencing (b) of *waal* in *E. coli*, *P. aeruginosa*, and *K. pneumoniae* strains.

- Gram negative bacteria are sensitive to complement-mediated killing as the MAC complex can insert into the outer membrane. Gram positive bacteria however are

resistant to complement mediated killing as the peptidoglycan of their membrane is too thick for MAC insertion. Thus,

5) In figure 8A All Staph aureus strain (Gram positive) are killing by the addition of glycine to serum. By what mechanism is this killing happening? Was a HI-serum (or C3 depleted) + glycine control used in these assays? In fact – was this control done for any of the strains in 8A?

Response, We appreciate this comment! According to Reviewer’s comments, we detected C3 and C5 on the surface of MRSA strains. These results have been described in lines 629 – 632 as following “Notably, MRSA is a Gram-positive bacterium deficient of outer membrane targeted by the MAC complex. Our results showed that higher level of C3 and C5 on the surface when glycine was supplemented (Supplementary Fig. 15), which requires for further study.”.

Supplementary Fig. 15 Flow cytometry quantification of anti-C3 and anti-C5 on the outer membrane surface of the indicated MRSA strains.

Meanwhile, a HI-serum + glycine control used in these assays has been added in Fig 8A following as:

Fig. 8a Percent survival of multidrug-resistant *E. coli*, *P. aeruginosa*, *K. pneumoniae*, and MRSA in the presence of 100 mM glycine, 100 μ L serum or both, or 100 μ L HI serum plus 100 mM glycine.

Minor comments:

1) In the introduction there is no mention of the role of capsule in serum resistance (although reports differ). Also no mention of antibody mediated serum resistance found in Gram negatives such as Salmonella (MacLennan et al. Science 2010), Pseudomonas (Wells et al 2014 JEM) and E. coli (Coggon et al. 2018 MBio).

Response, We appreciate this comment! According to Reviewer’s comments, they have been complemented in lines 61 - 77 as following “Most studies attribute the serum resistance of Gram-negative bacteria to bacterial outer membrane proteins, lipopolysaccharide (LPS) and its antibodies, and capsular polysaccharide in inhibiting complement system activation^{3, 6-11}. Outer membrane proteins bind C4b-binding

protein (C4BP), an endogenous complement-inhibiting factor, and factor H, which inhibits lectin-dependent cell killing¹¹⁻¹³. Serum-resistant strains also promote deposition of complement components at sites distant from the outer membrane, thereby preventing the efficient formation of membrane attack complexes (MACs), and preserving bacterial membrane integrity^{8, 14}. While OmpA binds C4BP and inhibits MAC attack^{12, 15, 16}, OmpC also plays a role because certain OmpC mutations increase bacterial survival in serum by inhibiting classical C1q and anti-OmpC antibody-dependent cell killing¹⁷. The serum resistome of *E. coli* EC958 comprised of 56 genes, where a major proportion of the genes encode membrane proteins or factors involving in LPS biosynthesis¹⁸. LPS is necessary for the survival of *E. coli* in the blood¹⁹. Immunoglobulin G to LPS confers serum resistance through shielding the bacterium from other antibodies that can induce complement-mediated killing of bacteria in *Salmonella*, *Pseudomonas aeruginosa* and *E. coli*^{10, 20, 21}. The presence of capsular polysaccharide is correlated with serum resistance in bacteria including *E. coli*^{11, 22}.

2) Fig 2b- no mention of what strain is used and text suggests could be Y17

Response, We appreciate this comment! It has been added in Figure legend. It is *E. coli* K12.

3) In many places I am not sure if stats are just missing or there was no significant difference even though graph and error bars look different; Fig 3d, Fig 8b, Sup Fig 5A,

Sup Fig 14

Response, We appreciate this comment! According to Reviewer's comments, they have been updated except for Supplementary Fig. 14, which has been removed according to Reviewer I suggestion.

4) Supplementary Figure 5A would be easier to interpret if the glycine minus results (in figure 4) were also put into the same graph. Could these figures be combined?

Response, We appreciate this comment! According to Reviewer's comments, Supplementary Figure 5A has been combined with supplementary Fig. 4B to Supplementary Figure 5A as following.

Supplementary Fig. 5 Glycine-enabled killing by serum. a, Percent survival of *E. coli* K12, Y5, Y10 and Y17 in the presence or absence of the indicated volume of serum or HI serum plus 100 mM glycine.

REVIEWERS' COMMENTS:

Reviewer #1 (Remarks to the Author):

All critiques have been satisfactorily addressed. Congratulations on this exciting study.

Jim Cassat

Reviewer #2 (Remarks to the Author):

In this re-submission by Chen et al, they have responded to all the comments I raised in my initial review in a satisfactory manner. A brief summary below:

-The addition of the O157:H7 result, as well as the O-antigen characterization are an excellent addition to the paper and strengthen the findings

- The C3 and C5 results on MRSA strains are interesting but agree that further study on this is outside scope of this paper.

-The HI serum + glycine controls has also improved the veracity of the findings.

-All minor comments were responded to as requested

REVIEWERS' COMMENTS:

Reviewer #1 (Remarks to the Author):

All critiques have been satisfactorily addressed. Congratulations on this exciting study.

Response, We appreciate it. Thanks.

Reviewer #2 (Remarks to the Author):

In this re-submission by Cheng et al, they have responded to all the comments I raised in my initial review in a satisfactory manner. A brief summary below:

-The addition of the O157:H7 result, as well as the O-antigen characterization are an excellent addition to the paper and strengthen the findings

- The C3 and C5 results on MRSA strains are interesting but agree that further study on this is outside scope of this paper.

-The HI serum + glycine controls has also improved the veracity of the findings.

-All minor comments were responded to as requested

Response, We appreciate it. Thanks.